# Simultaneous quantification of mRNA and protein in single cells reveals post-transcriptional effects of genetic variation

Christian Brion, Sheila M Lutz, Frank Wolfgang Albert*

Department of Genetics, Cell Biology and Development, University of Minnesota, Minneapolis, United States

**Abstract** *Trans*-acting DNA variants may specifically affect mRNA or protein levels of genes located throughout the genome. However, prior work compared *trans*-acting loci mapped in separate studies, many of which had limited statistical power. Here, we developed a CRISPR-based system for simultaneous quantification of mRNA and protein of a given gene via dual fluorescent reporters in single, live cells of the yeast *Saccharomyces cerevisiae*. In large populations of recombinant cells from a cross between two genetically divergent strains, we mapped 86 *trans*-acting loci affecting the expression of ten genes. Less than 20% of these loci had concordant effects on mRNA and protein of the same gene. Most loci influenced protein but not mRNA of a given gene. One locus harbored a premature stop variant in the *YAK1* kinase gene that had specific effects on protein or mRNA of dozens of genes. These results demonstrate complex, post-transcriptional genetic effects on gene expression.

## Introduction

Phenotypic variation in genetically complex traits is shaped by DNA variants throughout the genome. Many variants that influence complex traits alter gene expression (*Albert and Kruglyak, 2015*; *Maurano et al., 2012*). Understanding this regulatory genetic variation is critical for our knowledge of genetic disease, evolutionary change, and for applications of biology in agriculture and industry.

Regulatory genetic variation is typically studied by measuring mRNA abundance in genetically diverse populations to map regions of the genome that contain variants that cause variation in mRNA levels (expression quantitative trait loci, eQTLs). Many genes are influenced by *cis*-acting eQTLs that are caused by DNA variants in cis-regulatory elements. The proximity of *cis*-eQTLs to the genes they affect has aided their identification in numerous species (*GTEx Consortium, 2017*; *Albert et al., 2018*; *Brem et al., 2002*; *Cheung et al., 2005*; *Clément-Ziza et al., 2014*; *Hasin-Brumshtein et al., 2014*; *Higgins et al., 2018*; *Ka et al., 2013*; *Kita et al., 2017*; *Morley et al., 2004*; *West et al., 2007*). However, most genetic variation in gene expression arises from the combined effects of many *trans*-acting variants throughout the genome (*Albert et al., 2018*; *Grundberg et al., 2012*; *Wright et al., 2014*). *Trans*-acting variants affect the activity or abundance of diffusible factors that in turn alter the expression of other genes that can be located anywhere in the genome. Therefore, the identification of *trans*-eQTLs requires a large number of tests for statistical association between genotypes and mRNA levels. This, combined with the fact that *trans*-eQTLs have smaller effects than *cis*-eQTLs, has hampered the identification of individual *trans*-eQTLs in human association studies. By contrast, experimental crosses in organisms such as yeast (*Albert et al., 2018*; *Brem et al., 2002*; *Brion et al., 2020*; *Clément-Ziza et al., 2014*; *Emerson et al., 2010*; *Thompson and Cubillos, 2017*), plants (*Fu et al., 2013*; *West et al., 2007*; *Zhang et al., 2011*), worms (*Snoek et al., 2017*; *Viñuela et al., 2010*), flies (*Coolon et al., 2014*;

*For correspondence: falbert@umn.edu

Competing interests: The authors declare that no competing interests exist.

*Everett et al., 2020*; *Huang et al., 2015*; *McManus et al., 2010*), and mouse (*Gerrits et al., 2009*; *Hasin-Brumshtein et al., 2016* ) have quantified the global contribution of *cis* and *trans*-acting variation and identified thousands of *trans*-eQTLs.

Post-transcriptional mechanisms can shape protein levels independently of mRNA abundance (*Buccitelli and Selbach, 2020*; *Csárdi et al., 2015*; *McCarthy, 1998*), raising important questions about how DNA variation shapes protein levels compared to mRNA abundance. It is unclear whether most eQTLs, which by definition affect mRNA, also affect the protein abundance of the same genes, such that they also result in a protein-QTL (pQTL) (*Damerval et al., 1994*). It is also unclear to what extent there are protein-specific pQTLs that influence protein abundance without affecting the mRNA of the same gene. Recent studies have begun to address these questions (*Battle et al., 2015*; *Cenik et al., 2015*; *Chick et al., 2016*; *Foss et al., 2011*; *Foss et al., 2007*; *Ghazalpour et al., 2011*; *Großbach et al., 2019*; *Mirauta et al., 2020*; *Picotti et al., 2013*; *Sun et al., 2018*; *Wu et al., 2013*; *Yao et al., 2018*) and reported a wide range of observations.

Across several species, half or more of *cis*-eQTLs were reported to also affect protein abundance, suggesting high levels of agreement between mRNA and protein (*Albert et al., 2018*; *Albert et al., 2014b*; *Hause et al., 2014*; *Skelly et al., 2013*). However, other work found many *cis*-eQTLs whose effects on protein were smaller than those on mRNA, suggesting the existence of mechanisms that buffer protein levels against variation in mRNA abundance (*Battle et al., 2015*; *Ghazalpour et al., 2011*; *Mirauta et al., 2020*). *Trans*-acting QTLs were reported to show even greater differences between mRNA and protein. In mouse crosses and some yeast crosses, (*Mirauta et al., 2020*) *trans*-eQTLs and *trans*-pQTLs mapped for the same genes showed very little overlap, suggesting that while many *trans*-eQTLs are buffered at the protein level, there are also many *trans*-pQTLs that affect proteins via mechanisms that do not affect mRNA (*Chick et al., 2016*; *Foss et al., 2011*; *Foss et al., 2007*; *Ghazalpour et al., 2011*; *Williams et al., 2016*). While more recent work in yeast showed somewhat higher agreement between *trans*-eQTLs and *trans*-pQTLs (*Albert et al., 2018*; *Großbach et al., 2019*), yeast crosses also revealed instances of 'discordant' *trans*-acting loci that affect both mRNA and protein of the same gene, but in opposite directions (*Albert et al., 2018*; *Albert et al., 2014b*; *Foss et al., 2011*; *Großbach et al., 2019*). In plants, many *trans*-eQTLs appear to be buffered at the protein level, with few protein-specific pQTLs (*Fu et al., 2009*). In human populations, about half of the identified pQTLs could be accounted for by underlying eQTLs, although the exact fraction varied between studies (*Emilsson et al., 2018*; *Hause et al., 2014*; *Sun et al., 2018*; *Yao et al., 2018*). In yeast and humans, genetic effects on translation as measured by ribosome profiling were similar to those on mRNA, suggesting that translation cannot account for protein-specific pQTLs (*Albert et al., 2014a*; *Battle et al., 2015*; *Cenik et al., 2015*). Instead, protein-specific effects may arise from protein degradation, especially for proteins that are part of protein complexes (*Chick et al., 2016*; *Großbach et al., 2019*; *He et al., 2020*). Together, this literature suggests that genetic variants can independently affect the different layers of gene expression regulation.

However, there are substantial caveats to this conclusion. First, small sample sizes of dozens to a few hundred individuals were used in most studies to date. Most eQTLs and pQTLs change the expression of the gene they affect by 10% or less (*Albert et al., 2018*). Detecting such small-effect loci requires high statistical power resulting from the analysis of large numbers of individuals (*Albert et al., 2018*; *Albert et al., 2014b*; *Bloom et al., 2013*; *Ehrenreich et al., 2010*; *Visscher et al., 2017*). With limited sample size, real loci can be missed. For example, a locus that truly affects both mRNA and protein but happens to pass the detection threshold in mRNA but not in protein data could be incorrectly inferred to be buffered at the protein level. Thus, low statistical power can inflate discrepancies between eQTLs and pQTLs.

Second, experimental differences between studies may also inflate discrepancies between eQTLs and pQTLs. In nearly all cases, eQTLs and pQTLs were compared between experiments conducted in different laboratories, sometimes using different experimental designs, and, in the case of some human studies, comparing eQTLs and pQTLs mapped in different tissues. These comparisons are likely to be confounded by environmental influences, which can drastically alter the effects of regulatory variants. For example, stimulation of human immune cells can reveal new *cis*-eQTLs and *trans*-eQTLs (*Fairfax et al., 2014*; *Lee et al., 2014*). In yeast, many *trans*-eQTLs are specific to a specific culture medium (*Lewis et al., 2014*; *Smith and Kruglyak, 2008*), and some *trans*-eQTLs switch the sign of their effect in different media (*Smith and Kruglyak, 2008*). Even when experiments aim to

recreate the same conditions, subtle differences (for example, in the precise stage of cell growth, medium composition, temperature, or handling of the isolated mRNA or proteins) could influence measures of gene expression (*Gallego Romero et al., 2014*; *Gibson et al., 2007*; *Hayeshi et al., 2008*), potentially inflating the discrepancy between eQTLs and pQTLs.

To date, no study of regulatory variation has measured mRNA and protein with high statistical power and in the same samples. This challenge is not easy to overcome, because combined application of RNA-sequencing (RNA-seq) and mass spectrometry in thousands of matched samples remains prohibitively expensive. While recent studies in yeast (*Großbach et al., 2019*) mice (*Chick et al., 2016*; *Williams et al., 2016*), and human iPSC lines (*Mirauta et al., 2020*) used RNA-seq and mass spectrometry on the same cultures or tissue samples, their sample sizes of less than 200 individuals limited QTL detection. As a result, the role of genetic variation on post-transcriptional processes remains unclear, especially for *trans*-acting variation.

Here, we addressed this challenge by developing a system for quantifying mRNA and protein from the same gene simultaneously, in the same, live, single yeast cells using two fluorescent reporters. We reasoned that such an approach would equalize all environmental confounders and most of the technical biases that could obscure the relationship between eQTLs and pQTLs. Our assay is sensitive enough to be used in fluorescence-activated cell sorting (FACS), permitting the use of a well-powered bulk segregant QTL mapping approach (*Michelmore et al., 1991*) that leverages millions of genetically different yeast cells to yield high statistical power (*Albert et al., 2014b*; *Parts et al., 2014*). Genetic mapping across ten genes revealed 86 *trans*-acting loci. Although our approach was explicitly designed to maximize the chance of detecting similar genetic effects on mRNA and protein, the majority of the identified loci affected only mRNA or protein for a given gene, or had discordant effects on mRNA and protein. These results demonstrate considerable differences in the genetic basis of variation in mRNA vs protein abundance.

## Results

### A reporter system for quantifying mRNA and protein in single, live cells

We designed a dual reporter system for the simultaneous quantification of mRNA production and protein abundance of a given gene in single, live cells. In this system, protein abundance is measured via a fluorescent green fluorescent protein (GFP) tag fused to the C-terminus of the given protein of interest (*Huh et al., 2003*). To measure mRNA, we reasoned that a clustered regularly interspaced short palindromic repeats (CRISPR) guide RNA (gRNA) (*Doudna and Charpentier, 2014*) produced in equal molarity with the mRNA of interest would be able to drive proportional expression of a reporter gene via CRISPR activation (*Gilbert et al., 2014*; *Konermann et al., 2015*). To implement this idea, we created a gRNA tag located in the 3'UTR of the gene, downstream of the sequence encoding GFP (*Figure 1A*). After transcription of the mRNA along with this tag, the gRNA is released from the mRNA by two flanking self-cleaving ribozymes (Hammerhead, Hh; and Hepatitis Delta Virus, HDV) (*Gao and Zhao, 2014*). Because gRNA cleavage separates the mRNA from its poly-adenylated (polyA) tail, we added a synthetic polyA tail between the GFP tag and the Hh ribozyme (*Gao and Zhao, 2014*). Once released, the gRNA directs a catalytically deactivated CRISPR associated enzyme (dCas9) fused to a VP64 activation domain (dCas9-VP64) to drive the expression of a red fluorescent *mCherry* gene integrated in the genome (*Farzadfard et al., 2013*). After gRNA release, stability and half-life of the mRNA no longer affects gRNA abundance, such that mCherry expression primarily reports on mRNA production.

The reporter system is implemented as two cassettes (*Figure 1A*). The 'GFP-gRNA tag' cassette is added at the 3' end of the gene of interest. A second cassette, which we call the 'CRISPR reporter', comprises the remaining components: *dCAS9-VP64* and the *mCherry* gene under the control of an inactive *CYC1* promoter fragment. This promoter contains one recognition sequence that, when targeted by the gRNA and dCas9-VP64, drives mCherry expression (*Farzadfard et al., 2013*). The two cassettes are stored on two plasmids that can be used to easily construct strains for quantification of mRNA and protein of any gene of interest (*Figure 1—figure supplement 1*).

We tested the reporter system in diploid BY strains tagged at two genes with different expression levels: the highly expressed *TDH3*, and *GPD1*, which has an average expression level compared to other genes in the genome. Both genes gave green and red fluorescent signals in a plate-reader

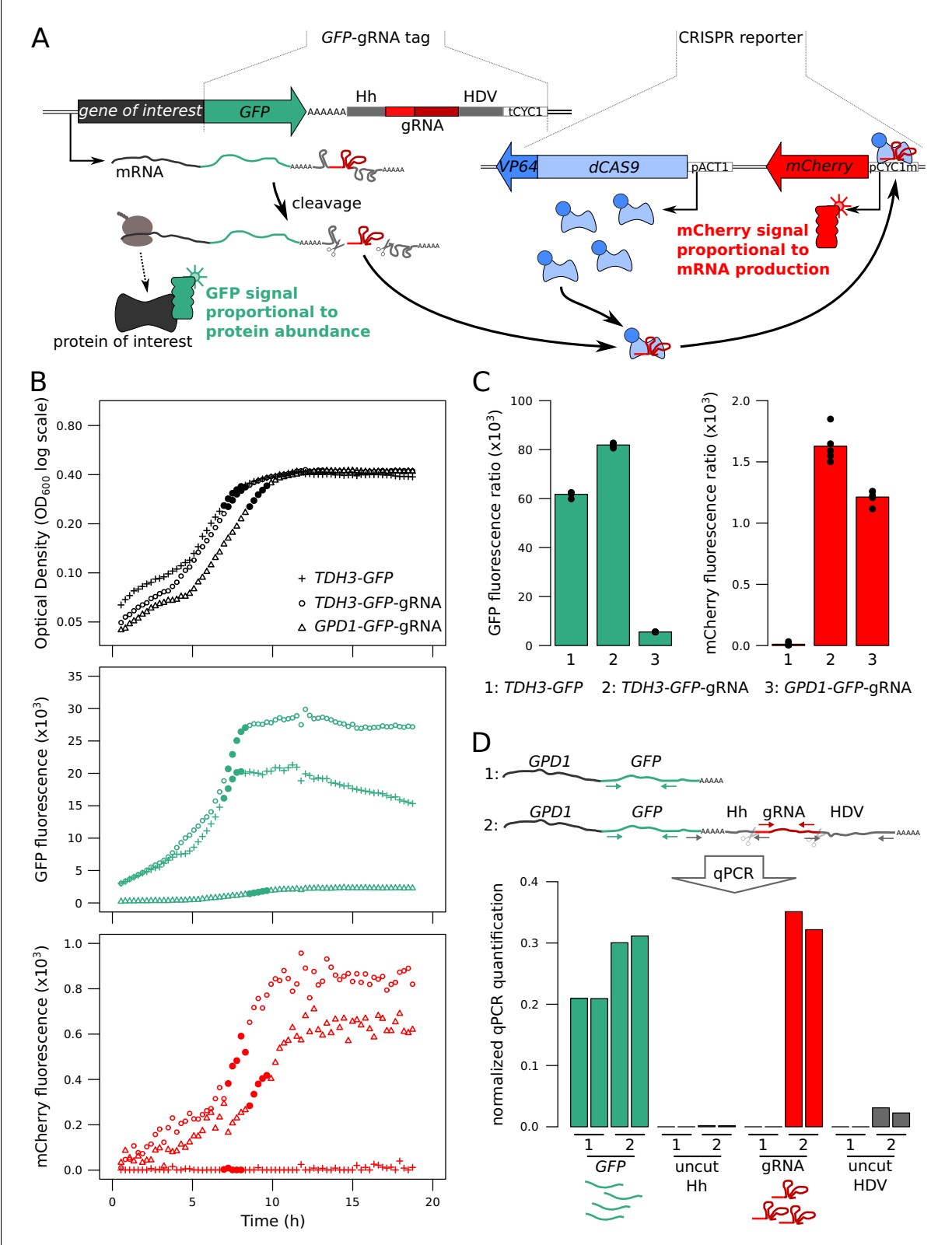

**Figure 1.** Fluorescence-based quantification of mRNA and protein levels. (**A**) Schematic of the dual quantification reporter. Hh: Hammerhead ribozyme, HDV: Hepatitis Delta Virus ribozyme, tCYC1: terminator sequence from the *CYC1* gene, *VP64*: four consecutive sequences encoding viral protein transcription activators VP16, pACT1: promoter sequence from the *ACT1* gene, pCYC1m: modified promoter sequence from the *CYC1* gene without baseline transcriptional activity. Plasmids implementing the reporter are shown in ***Figure 1—figure supplement 1***. (**B**) Time courses of cell density and

*Figure 1 continued on next page*

*Figure 1 continued*

fluorescence measurements for three tagged strains during 20 hr of growth on a plate reader. Filled symbols correspond to five measurements at the end of the exponential growth phase that were used for calculating fluorescence ratios for strain comparisons in the same physiological context as shown in panel C. (C) Fluorescence ratios (fluorescence / $OD_{600}$) for the three strains shown in panel B. The points show the fluorescence ratios for the five measurements shown as filled circles in panel B. (D) RNA quantification of the individual components of the tag, for *TDH3-GFP* and *TDH3-GFP-gRNA* by RT-qPCR. The two bars per strain show biological replicates. Normalized qPCR quantifications were calculated separately for each primer pair based on calibration with known template DNA amounts (*Figure 1—figure supplement 2*). For these pilot experiments, cells were grown in YNB glutamate medium. For experiments in B – D, the CRISPR reporter was inserted at *NPR2*.

The online version of this article includes the following figure supplement(s) for figure 1:

**Figure supplement 1.** Plasmids carrying the two reporter cassettes (top: GFP-gRNA tag, middle: CRISPR reporter) and the $Z_3EV$ construct (bottom).

**Figure supplement 2.** qPCR calibration plot for the four primer pairs, using a range of known DNA concentrations.

(*Figure 1B*). A strain carrying the CRISPR reporter and *TDH3* tagged with GFP but no gRNA produced no mCherry fluorescence, demonstrating that the gRNA is required for driving mCherry expression (*Figure 1B & C*). Quantitative real-time reverse-transcription PCR (qPCR) confirmed expression of the gRNA and the mRNA (*Figure 1D*). Absence of qPCR signal from primers that spanned the ribozyme cut sites in cDNA confirmed that the ribozymes cleaved the mRNA (*Figure 1D* and *Figure 1—figure supplement 2*).

## mCherry fluorescence provides a quantitative readout of mRNA production

To characterize the quantitative response of our reporter system to a range of gene expression levels, we used the synthetic $Z_3EV$ system, which allows quantitative regulation of transcription via the concentration of estradiol in the culture medium (*McIsaac et al., 2013*). We cloned the $Z_3EV$ promoter upstream of a *GFP*-gRNA sequence (*Figure 2A*) in a strain that also contained the CRISPR reporter and grew this strain in a range of estradiol concentrations. Along with the expected increase in green fluorescence (*McIsaac et al., 2013*), red fluorescence increased as a monotonic function of estradiol concentration (*Figure 2B*). Similar results were observed in the RM11-1a strain, which has a different genetic background than BY (*Figure 2—figure supplement 1*). Thus, mCherry provides a quantitative readout of the expression of the tagged gene.

While green fluorescence continued to increase throughout the tested estradiol range, red fluorescence ceased to increase at concentrations of more than 4 nM estradiol (*Figure 2B*). qPCR quantification of the gRNA showed that mCherry fluorescence followed gRNA abundance (*Figure 2C*), confirming that the mCherry reporter gene is quantitatively regulated by gRNA abundance. gRNA abundance was linearly related to *GFP* mRNA and GFP fluorescence at lower doses of estradiol but stopped increasing at higher doses (*Figure 2D & E*). This suggests that mCherry production is limited by gRNA availability at high expression levels. Increasing the concentration of dCas9 proteins or binding sites for the gRNA had no effect on the mCherry expression plateau (*Figure 2—figure supplement 2* and *Figure 2—figure supplement 3*).

The linear relationship between mCherry fluorescence and mRNA abundance of the tagged gene was present up to an expression level that corresponded to half of the abundance of *ACT1* mRNA, which we had used as a reference gene in qPCR (*Figure 2D*). In previous RNA-seq data (*Albert et al., 2018*), 95% of *S. cerevisiae* genes fall below this threshold (*Figure 2—figure supplement 4*, *Supplementary file 1*). While differences in growth media (SC here and YNB in the RNA-seq data), as well as different qPCR efficiencies for *ACT1* compared to GFP RNA (*Figure 1—figure supplement 2*), make this comparison imperfect, it suggests that the majority of genes can be quantified by our mRNA reporter. For lowly expressed genes, the GFP tag does not provide a strong enough signal to enable protein quantification (*Huh et al., 2003*; *Newman et al., 2006*, *Figure 2—figure supplement 4*). Based on these results, we concluded that our dual reporter system can be used to simultaneously measure mRNA and protein of more than half of the genes in the *S. cerevisiae* genome.

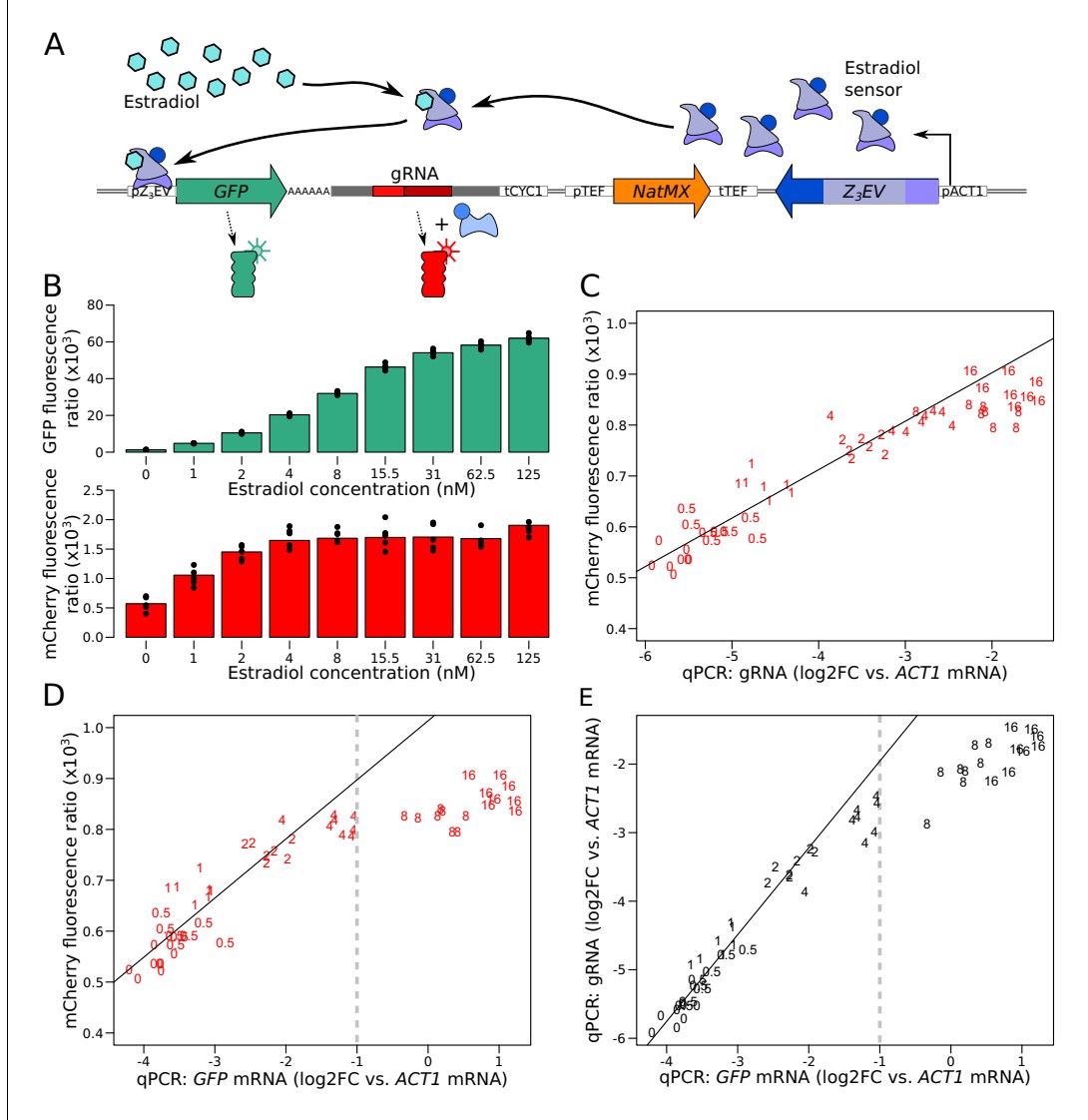

**Figure 2.** Characterization of the quantitative response of the reporter using an inducible transcriptional activator system to tune the expression of a *GFP* gene tagged with the gRNA. (**A**) Schematic of the $Z_3EV$ system used in this experiment. (**B**) Fluorescence as a function of increasing estradiol concentrations. (**C**) Comparison of gRNA abundance (qPCR) and mCherry fluorescence in increasing estradiol concentrations. qPCR quantifications were normalized across samples using *ACT1* cDNA as a reference. log2FC: $\log_2$ of fold-change (**D**) Comparison of mRNA abundance (qPCR) and mCherry fluorescence in increasing estradiol concentrations. (**E**) Comparison of mRNA abundance to gRNA abundance in increasing estradiol concentrations. The numbers in C to E show the concentration of estradiol in mM, with 7 to 8 biological replicates per concentration. Solid lines represent linear regressions calculated on measurements taken at up to 2 mM estradiol. Dashed vertical lines correspond to the mRNA abundance threshold, below which we deemed the reporter to be quantitative. Cells were grown in SC medium. See also *Figure 2—figure supplement 1*, *Figure 2—figure supplement 2*, *Figure 2—figure supplement 3*, *Figure 2—figure supplement 4*.

The online version of this article includes the following figure supplement(s) for figure 2:

**Figure supplement 1.** Quantitative response of the reporter measured using the $Z_3EV$ system in an RM strain.

**Figure supplement 2.** mCherry and GFP expression as a function of additional reporter component copies in a range of estradiol concentrations.

**Figure supplement 3.** Maps of the four plasmids used to investigate reporter behavior.

**Figure supplement 4.** Expression ranges of *S. cerevisiae* genes.

# Simultaneous mapping of genetic variation affecting mRNA and protein levels

Our reporter system quantifies mRNA production and protein abundance at the same time, in the same live cells, exposed to the same environment. These features enable mapping of the genetic basis for variation in mRNA and protein levels, free from environmental or experimental confounders. We selected ten genes for genetic mapping (*Figure 5—source data 1*), based on several criteria. Five genes (*ARO8*, *BMH2*, *GPD1*, *MTD1*, *UGP1*) had previously been reported to have multiple differences between their respective eQTLs (*Albert et al., 2018*) and pQTLs (*Albert et al., 2014b*). Three genes (*CYC1*, *OLE1*, *TPO1*) had shown high agreement between their eQTLs and pQTLs. The remaining two genes (*CTS1* and *RPS10A*) had low protein abundance based on GFP-tag quantification (*Huh et al., 2003*) compared to their mRNA levels (*Albert et al., 2018*).

To identify genetic loci affecting mRNA production and protein abundance, we used the strains BY4741 (BY), a reference strain frequently used in laboratory experiments, and RM11-1a (RM), a vineyard isolate closely related to European strains used in wine-making. These two strains differ at 47,754 variants in the yeast genome. We engineered RM to carry the CRISPR reporter inserted at the *NPR2* gene and a synthetic genetic array (SGA) marker for selection of *MAT*a haploid strains (*Tong and Boone, 2007*) at the neighboring *CAN1* gene. We engineered a series of BY strains, each carrying one gene tagged with the GFP-gRNA tag (*Figure 3*). We crossed these BY strains to the RM strain and obtained populations of recombinant haploid progeny carrying both the tagged gene and the CRISPR reporter. Flow cytometry detected a range of GFP and mCherry signals from single cells (*Figure 3*).

To study the relationship between mRNA and protein among single cells, we examined the cell-to-cell correlation between mCherry and GFP fluorescence in our genetically heterogeneous populations during exponential growth (*Figure 3—figure supplement 1A*). After correcting for cell size (*Figure 3—figure supplement 2*), mCherry and GFP were positively correlated for all tested genes (*Figure 3—figure supplement 1B*). The strength of the correlation varied from gene to gene. Lower correlations between mCherry and GFP were observed for the genes with many differences between published eQTLs and pQTLs compared to those with more concordant eQTLs and pQTLs. Thus, different genes are influenced by mRNA-specific or protein-specific variation to different degrees.

Fluorescence-based readouts of mRNA and protein quantification in single cells enabled the use of bulk segregant analysis, a genetic mapping paradigm that gains statistical power from the analysis of millions of cells (*Albert et al., 2014b*; *Ehrenreich et al., 2010*). In each of the segregating populations, we used FACS to collect four subpopulations of 10,000 cells with high or low GFP or mCherry fluorescence, respectively, at a cutoff of 3–5% (*Figure 3*). In prior work, similarly stringent selection provided high power for QTL mapping (*Albert et al., 2014b*; *Ehrenreich et al., 2010*; *Metzger and Wittkopp, 2019*; *Parts et al., 2014*).

To gauge the heritability of gene expression among single cells, we measured fluorescence between the high and low populations after 13 generations of growth. In almost all cases, the sorted populations showed significant (T-test, $p<10^{-5}$) differences in fluorescence, confirming the presence of genetic variants affecting mRNA and protein levels (*Figure 3—figure supplement 3*).

To map QTLs, we performed pooled whole-genome sequencing of all collected populations, computed the allele frequency of each DNA variant in each population, and calculated the difference in allele frequency (ΔAF) between high and low populations along the genome. A significant ΔAF at a locus indicated the presence of one or more genetic variants affecting protein abundance (GFP) or mRNA production (mCherry, *Figure 4—source data 1*). We call loci mapped in this paper 'mRNA-QTLs' and 'protein-QTLs', to distinguish them from published 'eQTLs' and 'pQTLs'. QTL mapping was performed in two to six biological replicates for all but one gene (*RPS10A*). Because any allele frequency differences among replicate populations sorted on the same parameters (e.g. two high GFP populations for the same gene) represent false positives, we used the replicate data to estimate false discovery rates. We chose a significance threshold (logarithms of the odds; 'LOD'=4.5) corresponding to a false discovery rate of 7% (*Figure 4—figure supplement 1*). Between replicates, 76% of the protein-QTLs and 78% of the mRNA-QTLs were reproducible at genome-wide significance (*Figure 4A*).

Across the ten genes, we detected 78 protein-QTLs and 44 mRNA-QTLs (*Figure 4—source data 2* and *Figure 4—source data 3*). By design, all detected loci were *trans*-acting, and most were

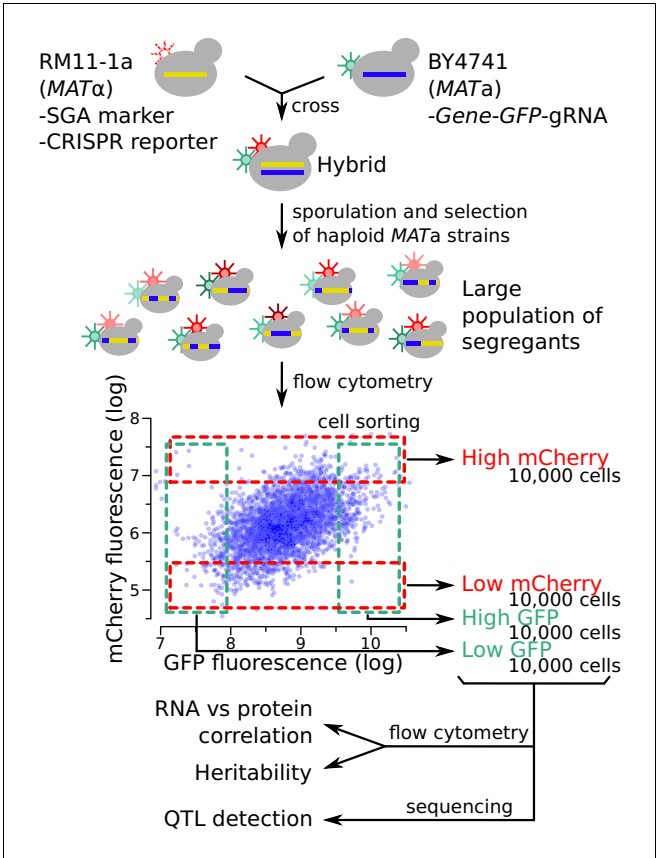

**Figure 3.** Schematic of the workflow for the identification of RNA-QTLs and protein-QTLs. The SGA marker allows for the selection of haploid *MAT*a strains after sporulation (Methods). The flow cytometry plot shows data transformed using the natural log. See also *Figure 3—figure supplement 1*, *Figure 3—figure supplement 2* & *Figure 3—figure supplement 3*.

The online version of this article includes the following figure supplement(s) for figure 3:

**Figure supplement 1.** mCherry and GFP fluorescence correlations.

**Figure supplement 2.** Correction of fluorescence measures for cell size to prevent spurious correlations between mCherry and GFP due to shared correlations with cell size.

**Figure supplement 3.** Heritability of mCherry and GFP fluorescence.

---

located on a different chromosome than the tagged gene. One locus located at ~450 kb on chromosome XIV affected mCherry levels in the same direction in all ten genes. This region was also observed in a control experiment, in which mCherry was expressed constitutively using an *ACT1* promoter, and without a gRNA present (*Figure 4—figure supplement 2*). This region harbors the *MKT1* gene, which carries a variant affecting a variety of traits (*Deutschbauer and Davis, 2005*; *Fay, 2013*). While the highly pleiotropic *MKT1* locus may truly affect all ten genes we tested, it could also affect mCherry fluorescence via mCherry maturation or degradation, independently of any tagged gene. We excluded this region from further analyses.

The number of protein-QTLs per gene identified here (median = 7) agrees well with results from a previous study using the same mapping strategy (median = 8 for the same genes; *Albert et al., 2014b*), confirming that individual proteins are influenced by multiple *trans*-acting loci. The effects of individual protein-QTLs and pQTLs showed a positive correlation across studies (Pearson r = 0.73, p-value<$10^{-15}$, *Figure 4B*). The number of mRNA-QTLs per gene in our study (median = 3 after removing the *MKT1* locus) was lower than th from a previous study using RNA sequencing in 1012 segregants (median = 8 for the same genes; *Albert et al., 2018*). This difference could be due to using our reporter in single cells with high stochastic variation compared to RNA-seq in individually grown segregant cultures in the earlier study (see Discussion). However, while the mRNA-QTLs

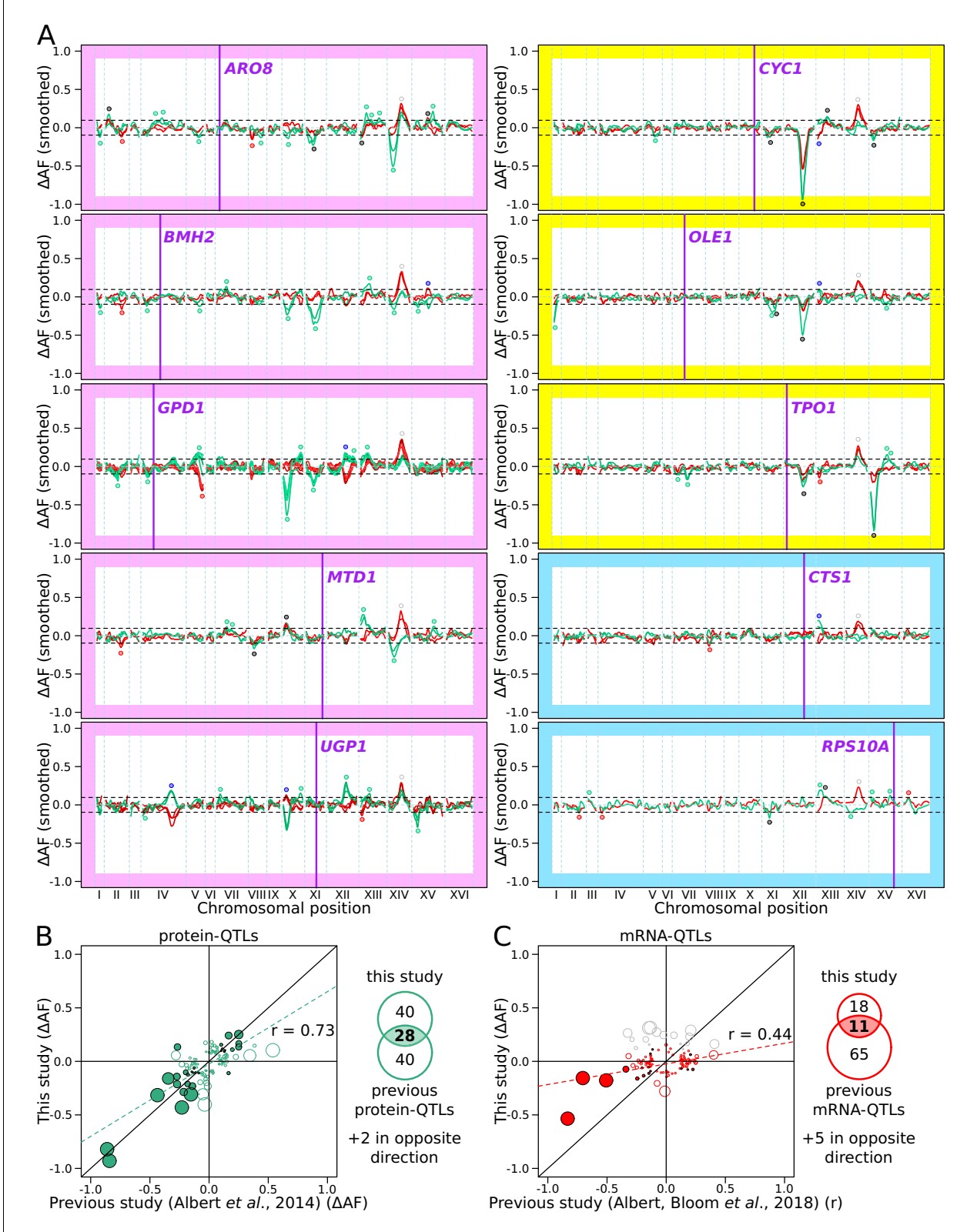

**Figure 4.** RNA-QTLs and protein-QTLs. (**A**) Allele frequency difference along the genome between the high and low population for each of the ten tagged genes, with 1–6 replicates per gene. Red and green curves correspond to the populations sorted to map mRNA and protein levels, respectively. The colored plot borders indicate the reason for which the gene was chosen for study. Pink: high discrepancy between reported eQTLs and pQTLs, yellow: high similarity between reported eQTLs and pQTLs, blue: high mRNA level and low GFP fluorescence. Purple vertical lines indicate

*Figure 4 continued on next page*

*Figure 4 continued*

the position of the tagged gene in the genome. Points indicate the location of significant QTLs, color coded based on protein or mRNA specificity (black: shared effect in same direction, green: protein-specific, red: mRNA-specific, blue: discordant). (B) Comparison between the effect size (ΔAF) of protein-QTLs identified in this study and pQTLs from previous work. (C) Same as (B) but comparing mRNA-QTLs identified in this study (ΔAF) to *trans*-eQTLs from previous work (eQTL effect sizes from *Albert et al., 2018* are shown as a Pearson correlation coefficient between mRNA abundance and genotype at the QTL marker). Filled circles correspond to QTLs significant in both datasets. Empty circles correspond to QTLs significant in only one dataset. Gray circles correspond to QTLs located on chromosome XIV between 350 and 550 kb, which were excluded from analysis. Circle size is proportional to the LOD score of the QTL. The Venn diagrams show the total number and overlap of QTLs detected across the 10 genes between studies. See also *Figure 4—figure supplement 1* & *Figure 4—figure supplement 2*. Source data files: *Figure 4—source data 1*, *Figure 4—source data 2*, *Figure 4—source data 3*.

The online version of this article includes the following source data and figure supplement(s) for figure 4:

**Source data 1.** List of all detected QTLs (LOD score >3.0) before combining replicates.
**Source data 2.** Comparison of significant protein-QTLs (LOD score >4.5) between this study and previous work (*Albert et al., 2014a*; used in *Figure 4B*).
**Source data 3.** Comparison of significant mRNA-QTLs (LOD score >4.5) between this study and previous work (*Albert et al., 2018* used in *Figure 4C*).
**Figure supplement 1.** Estimation of the false discovery rate (FDR) from comparisons across replicated sort experiments.
**Figure supplement 2.** Control mapping experiment, in which no gene was tagged with GFP or the gRNA, and in which the *mCherry* gene was under control of a constitutively active pACT1 promoter sequence.

detected by our reporter primarily reflect influences on mRNA production, the RNA-seq-based eQTLs may reflect effects on transcription as well as mRNA degradation, which our system was not designed to capture. The effects of mRNA-QTLs were significantly correlated between studies (r = 0.44, p-value=5 × 10$^{-6}$, *Figure 4C*). Some of the QTLs we detected harbored variants known to affect gene expression. For example, a region at ~650 kb on chromosome XII that contains the gene *HAP1* affected protein abundance and/or mRNA production of *GPD1*, *CYC1*, *OLE1*, and *TPO1* (*Figure 4A*). In the BY strain, the *HAP1* coding sequence is interrupted by a transposon insertion, which alters the expression of thousands of mRNAs in trans (*Albert et al., 2018*; *Brem et al., 2002*). Overall, these agreements with previous analyses confirmed the reliability of our new reporter as a means for mapping the genetic basis of gene expression variation.

We detected several QTLs that were not shared with prior work and *vice versa* (*Figure 4B-C*). Most of these QTLs tended to have small-effect sizes, suggesting that they could have been missed due to incomplete power in either study. Alternatively, these QTLs may reflect experimental differences between studies, such as different growth media. For example, we observed a new, strong protein-QTL affecting Aro8 on chromosome XIV. The regulation of Aro8 expression by amino acid levels (*Iraqui et al., 1998*) suggests that this QTL could be due to the synthetic complete medium used here vs. YNB medium in earlier work.

## Differences between mRNA-QTLs and protein-QTLs

Genetic mapping using our reporter enabled us to compare mRNA-QTLs and protein-QTLs, free from environmental or experimental confounders. We classified 86 loci based on the presence and effect direction of their respective mRNA-QTLs and/or protein-QTLs (*Figure 5A* and *Figure 5—figure supplement 1*, *Figure 5—source data 2*).

Of these 86 loci, 16 affected mRNA and protein of a given gene in the same direction. Such loci are expected for variants that alter a gene's mRNA production such that, in the absence of other effects, they also result in a concordant effect on protein abundance. A majority of the loci corresponded to protein-QTLs that did not overlap an mRNA-QTL. These 52 protein-specific QTLs may arise from variants that affect translation or protein degradation, without an effect on mRNA production.

There were eleven mRNA-QTLs that did not overlap with a protein-QTL and seven loci where mRNA-QTLs and protein-QTLs overlapped but had discordant effects. These two categories may occur when protein abundance and mRNA production of the same gene are regulated separately, through two different *trans*-acting pathways. These two pathways could be affected by two distinct but genetically linked causal variants at the same locus, or by a single variant with distinct pleiotropic effects on the two pathways. Alternatively, buffering mechanisms (*Battle et al., 2015*; *Großbach et al., 2019*) may compensate for changes in mRNA production perfectly (resulting in an

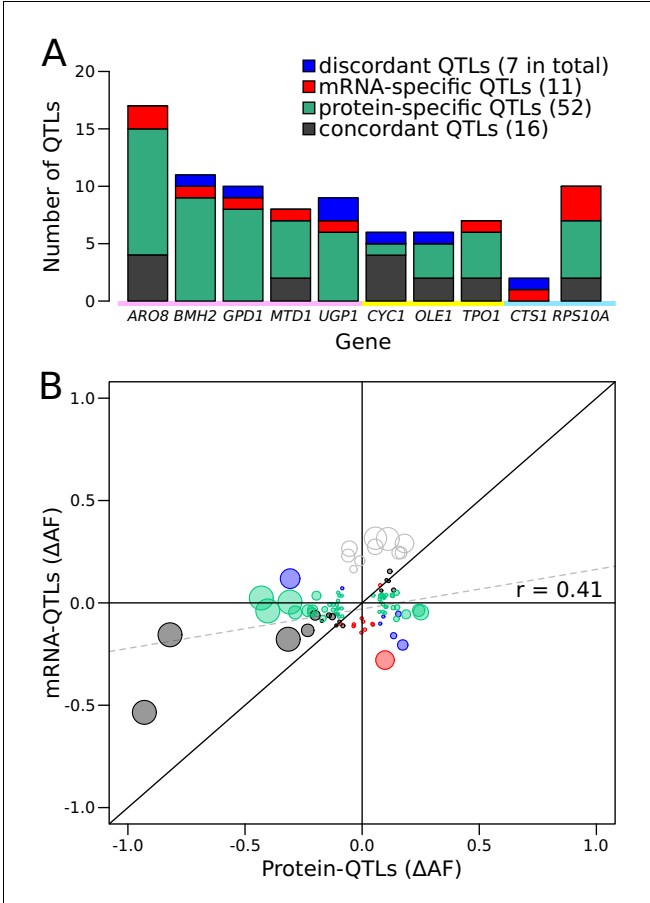

**Figure 5.** Comparison of RNA-QTLs and protein-QTLs. (A) Number of QTLs for each tagged gene, color coded according to type of effect on RNA and/or protein. (B) Comparison of QTL effect sizes between mRNA-QTLs and protein-QTLs. Gray circles correspond to QTLs located on chromosome XIV 350–550 kb, which were excluded from analysis. Circle size is proportional to the LOD score of the QTL. See also *Figure 5—figure supplement 1*. Source data files: *Figure 5—source data 1*, *Figure 5—source data 2*.

The online version of this article includes the following source data and figure supplement(s) for figure 5:

**Source data 1.** List of tagged genes and number of QTLs identified (used in *Figure 5B* and *6A*).
**Source data 2.** Comparison mRNA-QTLs and protein-QTLs (LOD score >4.5) (used in *Figure 5B*).
**Figure supplement 1.** Classification of QTLs into four groups based on their effect on protein and/or mRNA.

---

mRNA-specific QTL) or may overcompensate (resulting in a discordant QTL pair) (*Figure 5—figure supplement 1*).

Genes differed widely in the complexity and specificity of *trans*-acting loci that influenced their expression. For example, four genes (*BMH2*, *GPD1*, *UGP1*, and *CTS1*) were each influenced by multiple loci, none of which affected mRNA and protein levels in the same direction. By contrast, most of the loci influencing *CYC1* had concordant effects on mRNA and protein (*Figure 5A*).

While more than 73% of loci were specific for mRNA or protein, this difference might be inflated by loci that are truly concordant, but at which either the mRNA-QTL or the protein-QTL narrowly failed to meet the significance threshold. To bypass this potential limitation, we compared effect sizes, expressed as ΔAF, at significant mRNA-QTLs or protein-QTLs, irrespective of the significance of the locus in the other data (*Figure 5B*). When considering all loci, we observed a significant, positive correlation between mRNA and protein effect sizes (r = 0.41, p-value=$8.4 \times 10^{-5}$, *Figure 5B*). This overall correlation was almost exclusively driven by the concordant QTL pairs, whose effect sizes

showed a strong correlation (r = 0.88 p-value=$9 \times 10^{-6}$). In sharp contrast, neither protein-specific QTLs (r = 0.2, p-value=0.23) nor mRNA-specific QTLs (r = −0.05, p-value=0.9) had correlated effects across the two data types, as expected if these loci specifically affected only mRNA or only protein. Considerable differences between mRNA-QTLs and protein-QTLs were also observed when simply considering effect directions. Overall, only 64% of QTLs affected mRNA and protein in the same direction. While this was more than the 50% expected by chance (binomial test p-value=0.006), it left 36% of loci with discordant effects. Protein-specific QTLs showed similar directional agreement (63%) at lower significance (p-value=0.04), while only 55% of mRNA-specific QTLs had an effect in the same direction in the protein data, which was not significantly different from chance (p-value=0.5). Together, these results are consistent with the existence of many QTLs that specifically affect mRNA production or protein abundance.

Several loci were shared across the ten genes. Even these shared loci differed in the specificity of their effects on mRNA or protein. For example, the locus containing the *HAP1* gene had strong, concordant effects on both mRNA and protein for *CYC1* and *OLE1*, but affected only the protein abundance of *UGP1*, and had significant but discordant effects on mRNA and protein for *GPD1*. Overall, these results revealed complex *trans*-acting influences on gene expression, in which genes were affected by different sets of multiple loci, with different degrees of mRNA or protein specificity.

## A premature stop mutation in *YAK1* affects gene expression post-transcriptionally

The causal variants in most *trans*-acting loci are unknown, limiting our understanding of the underlying mechanisms. In particular, very few causal variants with specific *trans* effects on protein abundance are known (*Chick et al., 2016*; *Hause et al., 2014*). We noticed a region at ~155 kb on chromosome X that affected the protein abundance but not mRNA production of *ARO8*, *BMH2*, and especially *GPD1* (*Figure 4A*). This region spanned about 20 kb and contained 15 genes and 99 sequence variants. To identify the causal variant, we systematically divided this region into four tiles, swapped alleles in each tile using double-cut CRISPR-swap, an efficient scarless genome editing strategy (*Lutz et al., 2019*), and quantified the effect of these swaps on Gpd1-GFP fluorescence (*Figure 6A-D*).

This strategy, followed by analysis of our segregant population sequencing data, pinpointed a single G→A variant at 148,659 bp in the *YAK1* gene as the causal variant. While this variant is present in neither the BY nor RM reference genomes (*Figure 6E* and *Figure 6—figure supplement 1*), our sequence data showed it to be present in all BY derivatives we used from the GFP collection (specifically, strains tagged at *ARO8*, *BMH2*, *GPD1*, *MTD1*, and *UGP1*; *Figure 6—figure supplement 1*; *Huh et al., 2003*). We observed this variant in two additional strains we genotyped from the GFP collection (*FAA4* and YMR315W) and all four strains we genotyped from the tandem affinity purification (TAP)-tagged collection (*PGM1*, *NOT5*, *EMI2*, and *TUB1*) (*Ghaemmaghami et al., 2003*). This variant was not present in a BY4741 strain that we obtained from the ATCC stock center (#201388), suggesting that the *YAK1* variant arose very recently in the specific BY4741 strain used to construct both the GFP and TAP-tagged collections. *YAK1* encodes a protein kinase involved in signal transduction in response to starvation and stress, indirectly regulating the transcription of genes involved in various pathways (*Figure 6E*). The causal variant changes the 578th codon (glutamine) to a premature stop codon that is predicted to disrupt translation of the Yak1 kinase domain (*Figure 6E*).

The $YAK1^{Q578*}$ variant led to a diminution of Gpd1-GFP fluorescence, suggesting a decrease of Gpd1-GFP protein abundance (*Figure 6D*). While *YAK1* may control transcription of genes in the glycerol biosynthesis pathway (*Lee et al., 2008*; *Rep et al., 2000*), which includes *GPD1*, our QTL results suggested no link between the variant and *GPD1-GFP* mRNA level. Consistent with a protein-specific *trans*-effect on *GPD1*, deletion of *YAK1* in a strain in which *GPD1* was tagged with GFP-gRNA caused a reduction of green fluorescence but had no detectable effect on mCherry fluorescence (*Figure 6F*). Further, qPCR indicated no difference in the level of *GPD1-GFP* mRNA in $YAK1^{Q578*}$ or $yak1\Delta$ compared to matched $YAK1^{wt}$ (*Figure 6F*).

We explored the genome-wide effects of the *YAK1* variant by comparing mRNA and protein abundance between strains carrying the $YAK1^{Q578*}$ and the BY *YAK1* wildtype allele using RNA-seq and mass spectrometry (*Figure 7A*, *Figure 7—source data 1*, *Figure 7—source data 2* and *Figure 7—source data 3*). Among 5755 quantified mRNA transcripts, 262 were up-regulated and 310 down-regulated in the presence of $YAK1^{Q578*}$ (Benjamini-Hochberg (BH) adjusted p-value<0.05)

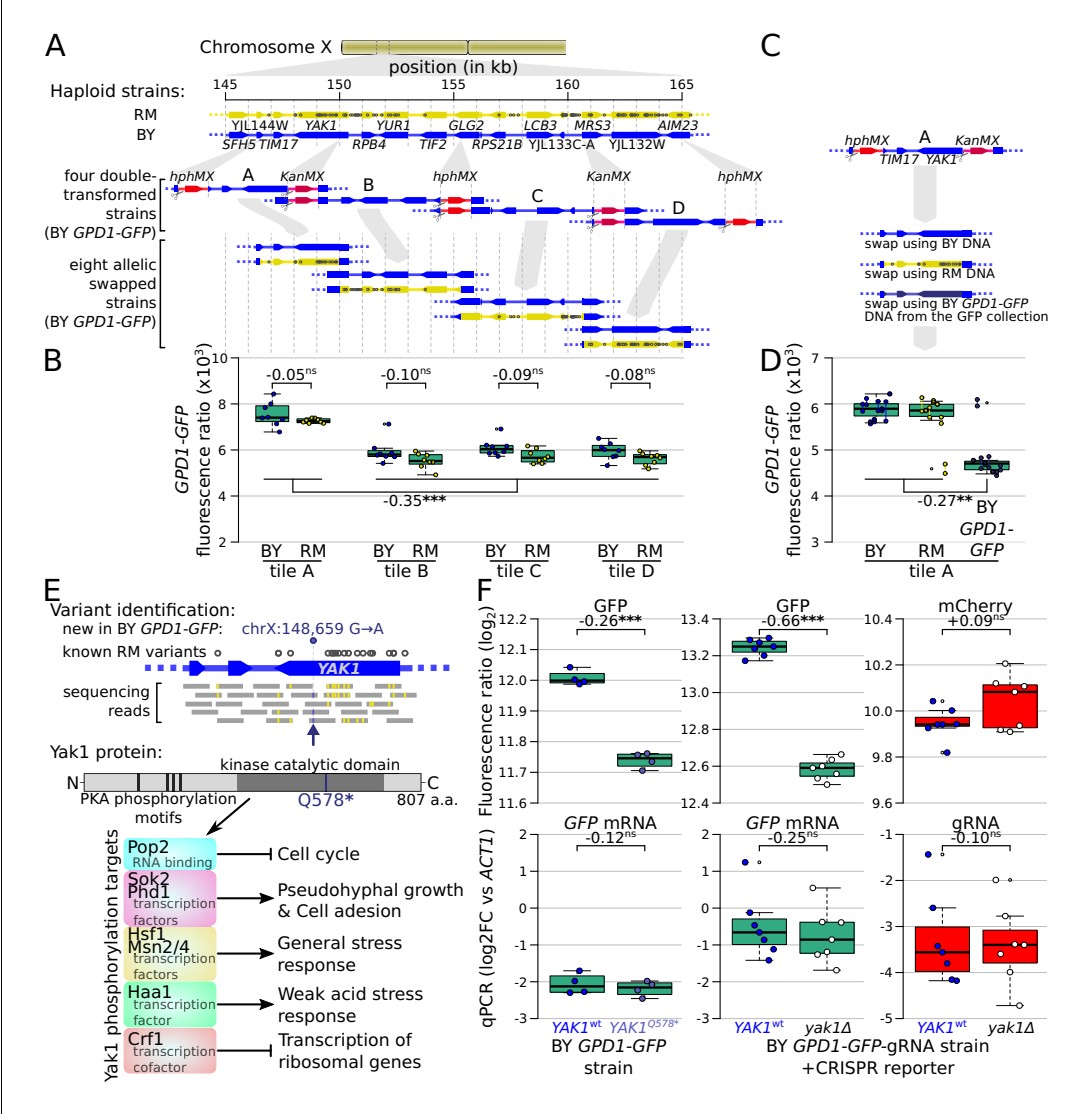

**Figure 6.** Identification of a causal variant influencing Gpd1-GFP protein but not *GPD1* mRNA. (**A**) Schematic of the investigated region and the strategy for generating tiled allele swaps across the region. Gray dots on the RM genome (yellow) indicate the positions of known BY/RM variants. (**B**) Boxplots comparing Gpd1-GFP fluorescence between allele swaps (6–8 replicates per swap). While none of the swaps resulted in a difference in fluorescence between BY and RM alleles, replacement with both backgrounds in tile A generated a significant increase in green fluorescence. Based on this result, we suspected that the BY *GPD1-GFP* strain carried a new mutation that was absent from the RM as well as BY genomic DNA used for the replacement. (**C**) Schematic of using BY *GPD1-GFP* DNA as a repair template for the allelic swap of tile A. (**D**) Boxplots comparing Gpd1-GFP fluorescence between the three swaps of tile A. The DNA repair template from the BY *GPD1-GFP* strain resulted in low Gpd1-GFP fluorescence, suggesting a new mutation in the BY *GPD1-GFP* strain. (**E**) Identification of the *YAK1*$^{Q578*}$ mutation using sequencing data from the segregant population, and location of *YAK1*$^{Q578*}$ in the Yak1 kinase protein sequence. Selected known protein phosphorylation targets of Yak1 and downstream processes are indicated. (**F**) Effect of *YAK1*$^{Q578*}$ and *YAK1* knockout on Gpd1-GFP expression. Top: fluorescence, bottom: RNA quantified by qPCR. Numbers atop the boxplots correspond to $\log_2$(fold-change). log2FC: $\log_2$ of fold-change. Stars indicate the significance of a t-test: ns: not significant ($p>0.05$); *: $0.005 < p < 0.05$; **: $0.0005 < p < 0.005$; ***: $p<0.0005$. Cells were grown in SC medium. See also *Figure 6—figure supplement 1* & *Figure 6—figure supplement 2*.

The online version of this article includes the following figure supplement(s) for figure 6:

**Figure supplement 1.** Allele frequency of variants in tile A for every sequenced segregant population, separated between populations based on BY strains from the GFP collection (top), or on a separate BY strain into which we had engineered the GFP tag (bottom).

**Figure supplement 2.** Effect of *YAK1*$^{Q578*}$ and *YAK1* deletion on growth rate and *GPD1* expression during chronic osmotic stress.

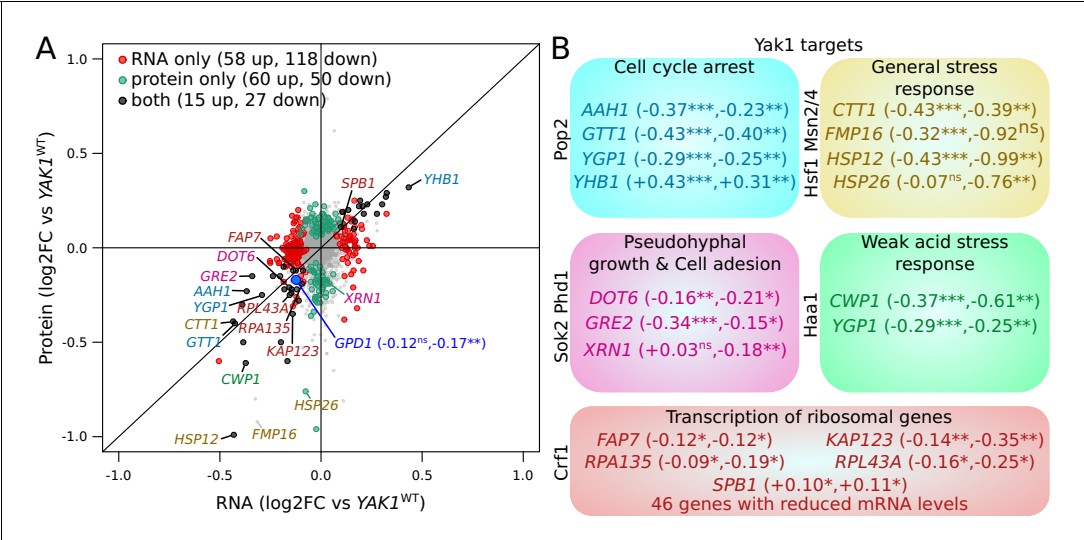

**Figure 7.** Effect of $YAK1^{Q578*}$ on gene expression. (A) Effect on mRNA levels and protein levels quantified by RNA sequencing and mass spectrometry, respectively. Genes are colored according to their function as indicated in (B). *GPD1* is highlighted in blue. (B) Examples of differentially expressed genes related to processes downstream of Yak1 phosphorylation regulation. The two numbers following gene names correspond to the $\log_2$ of fold-change (log2FC) of differential expression for mRNA and protein abundance, respectively. Stars indicate the significance of differential expression (Benjamini-Hochberg adjusted p-values). ns: not significant, p>0.05; *: 0.005 < p < 0.05; **: 0.0005 < p < 0.005; ***: p<0.0005. Cells were grown in SC medium. See also *Figure 7—figure supplement 1*. Source data files: *Figure 7—source data 1*, *Figure 7—source data 3*, *Figure 7—source data 4*, *Figure 7—source data 2*.

The online version of this article includes the following source data and figure supplement(s) for figure 7:

**Source data 1.** Differential expression analysis of RNA abundance of BY *YAK1*-Q578* compared to BY *YAK1*-WT.
**Source data 2.** GO term enrichment analysis of the differentially expressed genes (BY *YAK1*-Q578* compared to BY *YAK1*-WT).
**Source data 3.** Differential expression analysis of protein abundance of BY *YAK1*-Q578* compared to BY *YAK1*-WT.
**Source data 4.** Comparison of the differential expression of mRNA versus protein (BY *YAK1*-Q578* compared to BY *YAK1*-WT, used in *Figure 7* and *Figure 7—figure supplement 1*).
**Figure supplement 1.** $YAK1^{Q578*}$ effect on the expression of genes involved in cytoplasmic translation.

(*Benjamini and Hochberg, 1995*). The variant reduced the abundance of 82 of 2590 quantified proteins, and increased another 82 proteins (BH adjusted p-value<0.05). By comparing mass spectrometry and RNA-seq results, we classified genes as affected only at the mRNA level (58 genes up, and 118 genes down-regulated), only at the protein level (60 genes up, and 50 genes down-regulated), or at both mRNA and protein (15 genes up, and 27 genes down-regulated). There was a strong enrichment for genes involved in cytoplasmic translation (q-value $<10^{-10}$) among the genes with reduced mRNA abundance, which is consistent with the role of Yak1 as a regulator of transcription of ribosomal genes through Crf1 phosphorylation (*Martin et al., 2004*; *Figure 7—figure supplement 1*, *Figure 7—source data 2*). Genes up-regulated at the mRNA level showed an enrichment in amino acid biosynthesis (q-value = 0.001). The most differentially expressed genes included known targets of the *YAK1* pathway (*Figure 7A–B*). Gpd1 protein was strongly reduced (BH adjusted p-value<0.004), with a non-significant effect on *GPD1* mRNA (adjusted p-value=0.10) (*Figure 7A*).

Finally, we investigated if the $YAK1^{Q578*}$ mutation affects other phenotypes. As *YAK1* and *GPD1* are involved in osmotic stress resistance (*Lee et al., 2008*), we grew strains carrying $YAK1^{wt}$, $YAK1^{Q578*}$, and $yak1\Delta$, in a range of sodium chloride concentrations (*Figure 6—figure supplement 2A*). While this osmotic stress reduced growth, strains with $YAK1^{Q578*}$ and $yak1\Delta$ had a higher growth rate than wild-type, consistent with the role of Yak1 in triggering cell cycle arrest in response to stress. Gpd1-GFP abundance increased with stronger osmotic stress in $YAK1^{wt}$ and $yak1\Delta$, with consistently lower expression of Gpd1-GFP in $yak1\Delta$ (*Figure 6—figure supplement 2B–C*).

## Discussion

We developed a fluorescence-based dual reporter system for the simultaneous quantification of mRNA and protein from a given gene in single, live cells. This system enabled the use of a statistically powerful mapping strategy to identify genetic loci that affected mRNA production or protein abundance in *trans*. Because mRNA and protein were quantified in exactly the same condition, mRNA-QTLs and protein-QTLs could be compared without environmental confounding. This reporter system and mapping strategy were explicitly designed to maximize the chance of detecting concordant effects on mRNA and protein. Nevertheless, of the 86 *trans*-acting loci that contributed to variation in the expression of ten genes, 84% did not have concordant effects on mRNA production and protein abundance. This result demonstrates the importance of variants that act on specific layers of gene expression.

The genes *ARO8*, *BMH2*, *GPD1*, *MTD1*, and *UGP1*, which we had selected for high discrepancy between previous mRNA-QTLs and protein-QTLs, showed many QTLs (89%) that were not shared between mRNA and protein in our data. In contrast, *CYC1*, *OLE1*, and *TPO1*, which we had selected for higher agreement between published QTLs, showed fewer discrepant QTLs in our data, although even for these genes the majority of QTLs was not shared between mRNA and protein (58%). These three genes showed fewer QTLs overall and all had one locus with strong effect size (*Figure 4A*; the *HAP1* locus for *CYC1* and *OLE1*, and the *IRA2* locus (*Brem et al., 2002*; *Smith and Kruglyak, 2008*) for *TPO1*). Based on these results, strong *trans*-acting loci may be more likely to cause concordant effects on mRNA and protein, while loci with smaller effects could be more likely to be specific to mRNA or protein.

While half of the mRNA-QTLs we detected had concordant effects on protein (16 out of 34), most protein-QTLs had no effects on mRNA (52 out of 75), in line with observations from prior work (*Albert et al., 2014b*; *Albert et al., 2018*; *Foss et al., 2011*). That 70% of our protein-QTLs had protein-specific effects suggests that the causal variants underlying many of these loci affect post-transcriptional processes, *i.e.* mRNA stability, translation, or protein degradation. The indirect nature of our mCherry reporter and its lower signal intensity compared to GFP fluorescence are potential sources of measurement noise, which could have reduced detection power for mRNA-QTLs compared to protein-QTLs. However, our analyses of the magnitudes and directions of effects on mRNA and protein, which did not require loci in the other data to meet a significance threshold, also suggested that many protein-QTLs specifically influence protein.

We found seven loci that had discordant effects on mRNA production and protein abundance of the same gene. For example, at the *HAP1* locus, the BY allele increased Gpd1 protein abundance but decreased *GPD1* mRNA production, as had been seen when comparing QTLs across different studies (*Albert et al., 2018*). The highly pleiotropic effects of *HAP1* on mRNA and protein levels of many genes (*Albert et al., 2014b*; *Albert et al., 2018*; *Smith and Kruglyak, 2008*) reinforces the hypothesis that *HAP1* alleles influence Gpd1 protein abundance and mRNA production via two separate *trans*-acting mechanisms.

QTLs with discordant effects on mRNA and protein, as well as mRNA-specific QTLs, may be caused by buffering of mRNA variation at the protein level. A well-studied example of this phenomenon is the regulation of expression of genes that encode members of a protein complex, in which excess protein molecules that cannot be incorporated in the complex are rapidly degraded (*Taggart et al., 2020*). Among the genes we investigated, *RPS10A* encodes a part of the ribosome small subunit complex. *RPS10A* showed the highest number of mRNA-specific QTLs, possibly because Rps10a is subject to buffering mechanisms.

The nonsense mutation (Q578*) we identified in the *YAK1* gene provides an informative example of the complexity with which *trans*-acting variants can shape the transcriptome and the proteome. $YAK1^{Q578*}$ changed protein abundance for many genes more strongly than mRNA abundance, but also affected mRNA but not protein for many other genes. Thus, the consequences of this mutation span mechanisms that affect mRNA as well as protein-specific processes. A reduction of ribosomal gene transcription may account for some of these observations by reducing the translation of multiple genes.

The $YAK1^{Q578*}$ variant likely arose as a new mutation in the BY4741 ancestor of the GFP and TAP-tagged collections. Its relatively large effect and rarity in the global yeast population are consistent with population genetic expectations (*Eyre-Walker, 2010*; *Gibson, 2012*) and observations

(*Bloom et al., 2019*; *Fournier et al., 2019*) for a deleterious variant that may have drifted to fixation in this specific background, as has been suggested for many causal variants in natural yeast isolates (*Warringer et al., 2011*). Alternatively, faster growth of a strain carrying the $YAK1^{Q578*}$ variant during osmotic stress (*Figure 5—figure supplement 1*) may have contributed to adaptive fixation of this variant in this specific isolate of BY4741. While the large effect of $YAK1^{Q578*}$ aided our ability to fine-map it (*Rockman, 2012*), we suspect that its diverse, mRNA-specific as well as protein-specific mechanistic consequences may be representative of more common *trans*-acting variants with smaller effects.

To simultaneously quantify mRNA and protein and eliminate potential environmental confounders in expression QTL mapping, we developed a system in which a gRNA drives CRISPR activation of a fluorescent reporter gene in proportion to a given mRNA of interest. Standard methods for mRNA quantification require lysis of cell cultures or tissues, constraining sample throughput and statistical power for mapping regulatory variation. Single-cell RNA-seq (*Picelli, 2017*; *Tang et al., 2009*) and in situ fluorescent hybridization techniques (*Buxbaum et al., 2014*; *Player et al., 2001*; *Rouhanifard et al., 2018*) are improving rapidly, including in yeast (*Gasch et al., 2017*; *Li and Neuert, 2019*; *Nadal-Ribelles et al., 2019*; *Wadsworth et al., 2017*). However, these approaches still have throughput that is orders of magnitude below that available using FACS. Further, they involve cell lysis or fixation, precluding bulk segregant approaches that rely on growing cells after sorting. By contrast, our reporter allows quantification of mRNA production of a given gene within millions of single, live cells by flow cytometry. Because mCherry production in our system amplifies mRNA abundance signals, it is able to quantify genes with mRNA levels that would likely be hard to detect by FACS using in situ hybridization methods. The readout of our system is driven by a gRNA after it detaches from its mRNA. Therefore, the resulting signal is independent of the fate of the mRNA after gRNA release. Given the hammerhead ribozyme has a rate constant for self-cleavage of 1.5 per minute (*Wurmthaler et al., 2018*), gRNA abundance is not expected to reflect the half-life and stability of most yeast mRNAs, which have a median half-life of 3.6 min (*Chan et al., 2018*). By contrast, standard methods usually used in eQTL mapping quantify mRNA at steady state, which may explain some of the differences we observed between our mRNA-QTLs and known eQTLs identified by RNA-seq.

Future versions of our reporter could be improved in several ways. The mCherry used here has a maturation time of about 40 min (*Merzlyak et al., 2007*), which limits the temporal resolution at which we can observe dynamic expression changes. Fluorescent proteins with faster maturation times and shorter half-lives, could enable precise investigation of rapid temporal change in mRNA production. Using brighter fluorescent proteins or multiple copies of mCherry and its gRNA-targeted promoter could increase fluorescence and increase mRNA detection further. Finally, we observed that beyond a certain mRNA level, the abundance of gRNA no longer follows that of the tagged mRNA. Nevertheless, we estimated that our CRISPR-based reporter can be used to quantify the mRNA production of most *S. cerevisiae* genes. Like other reporters that tag the 3' end of genes, our system decouples the native 3' UTR from the gene of interest. 3' UTRs can influence gene expression (*Shalem et al., 2015*), and genetic *trans*-effects that involve mechanisms that target the 3' UTR would be missed by our system. However, loss of the native 3' UTR is not expected to inflate differences between mRNA-QTLs and protein-QTLs. Instead, 3' UTR removal may reduce differences between mRNA-QTLs and protein-QTLs because it eliminates an element of the mRNA that could be a target for mRNA-specific processes. Finally, CRISPR activation has been demonstrated in many organisms (*Long et al., 2015*; *Maeder et al., 2013*; *Park et al., 2017*), suggesting that similar reporters for RNA production could be developed in other species.

Our reporter system for quantifying mRNA and protein of a given gene in the same live, single cells combined with a mapping strategy with high statistical power was deliberately designed to minimize technical or environmental confounders that may have inflated differences between the genetics of mRNA and protein levels in earlier work. Yet, fewer than 20% of the detected *trans*-acting loci we detected had concordant effects on mRNA and protein levels, providing strong support for the existence of discordant *trans* effects on mRNA vs proteins. Whether *cis* effects show similar degrees of specificity will be a topic of future work. The fact that the majority of *trans*-QTLs identified here were protein-specific suggests that protein abundance is under more complex genetic control than mRNA abundance.

# Materials and methods

## Key resources table

| Reagent type (species) or resource | Designation | Source or reference | Identifiers | Additional information |
|---|---|---|---|---|
| Gene (*S. cerevisiae*) | *NPR2* | SGD | YEL062W | insertion site for CRISPR reporter |
| Gene (*S. cerevisiae*) | *TDH3* | SGD | YGR192C | tagged gene |
| Gene (*S. cerevisiae*) | *ARO8* | SGD | YGL202W | tagged gene |
| Gene (*S. cerevisiae*) | *BMH2* | SGD | YDR099W | tagged gene |
| Gene (*S. cerevisiae*) | *GPD1* | SGD | YDL022W | tagged gene |
| Gene (*S. cerevisiae*) | *MTD1* | SGD | YKR080W | tagged gene |
| Gene (*S. cerevisiae*) | *UGP1* | SGD | YKL035W | tagged gene |
| Gene (*S. cerevisiae*) | *CYC1* | SGD | YJR048W | tagged gene |
| Gene (*S. cerevisiae*) | *OLE1* | SGD | YGL055W | tagged gene |
| Gene (*S. cerevisiae*) | *TPO1* | SGD | YLL028W | tagged gene |
| Gene (*S. cerevisiae*) | *CTS1* | SGD | YLR286C | tagged gene |
| Gene (*S. cerevisiae*) | *RPS10A* | SGD | YOR293W | tagged gene |
| Gene (N.A) | *dCAS9* | Addgene 49013, **Farzadfard et al., 2013** (doi:10.1021/sb400081r) | | |
| Gene (N.A) | *mCherry* | Addgene 25444 | | |
| Gene (N.A) | *GFP* | **Huh et al., 2003** (doi:10.1038/nature02026) | | |
| Gene (N.A) | *Z3EV* | **McIsaac et al., 2013** (doi:10.1093/nar/gks1313) | | |
| Strain background (*S. cerevisiae*) | BY4741 | **Albert et al., 2018** (doi:10.7554/eLife.35471) | | |
| Strain background (*S. cerevisiae*) | RM11.1a | **Albert et al., 2018** (doi:10.7554/eLife.35471) | | |
| Strains (*S. cerevisiae*) | 167 *S. cerevisiae* strains | this paper | YFA- | **Supplementary file 2** |
| Recombinant DNA reagent | 21 Plasmids (Stored in *E. coli*) | Addgene/this paper | BFA- | **Supplementary file 4** |
| Sequence-based reagent | 143 Primers | this paper (IDT) | PFA- | **Supplementary file 3** |
| Commercial assay or kit | Phusion Hot Start Flex | NEB | M0535L | Cloning and genome editing |
| Commercial assay or kit | Monarch DNA Gel Extraction Kit | NEB | T1020L | Cloning and genome editing |
| Commercial assay or kit | Taq DNA Polymerase | NEB | M0267L | Cloning and genome editing |
| Commercial assay or kit | NEBuilder HiFi DNA Assembly Cloning Kit | NEB | E5520S | Cloning |

*Continued on next page*

*Continued*

| Reagent type (species) or resource | Designation | Source or reference | Identifiers | Additional information |
|---|---|---|---|---|
| Commercial assay or kit | ZR Quick-RNA Kit | Zymo Research | R1054 | qPCR |
| Commercial assay or kit | GoScript RT kit | Promega | A5000 | qPCR |
| Commercial assay or kit | GoTaq qPCR kit | Promega | A6001 | qPCR |
| Commercial assay or kit | E-Z-96 Tissue DNA kit | Omega | D1196-01 | WGS |
| Commercial assay or kit | Nextera DNA Library Prep kit | Illumina | FC-121–1030 | WGS |
| Commercial assay or kit | ZR Fungal/Bacterial RNA mini-prep kit | Zymo Research | R2014 | RNA-seq |
| Commercial assay or kit | NEBNext Poly(A) mRNA Magnetic Isolation Module | NEB | E7490L | RNA-seq |
| Commercial assay or kit | EB Ultra II Directional RNA library kit for Illumina | NEB | E7760 | RNA-seq |
| Commercial assay or kit | NEBNext Multiplex Oligos for Illumina | NEB | E7600S | RNA-seq |
| Chemical compound, drug | YNB media | VWR | 97064–162 | growth media |
| Chemical compound, drug | SC complement | Sunrise science | 1342–030 | growth media |
| Chemical compound, drug | Estradiol | Sigma-Aldrich | E1024-1G | Media complement |
| Software, algorithm | R version 3.5.1 | https://www.r-project.org | | Data analysis |
| Software, algorithm | BWA | *Li and Durbin, 2009* (doi:10.1093/bioinformatics/btp324) | | WGS |
| Software, algorithm | samtools | *Li and Durbin, 2009* (doi:10.1093/bioinformatics/btp352) | | WGS |
| Software, algorithm | MULTIPOOL | *Edwards and Gifford, 2012* (doi:10.1186/1471-2105-13-S6-S8) | | QTL analysis |
| Software, algorithm | trimmomatic | *Bolger et al., 2014* (doi:10.1093/bioinformatics/btu170) | | RNA-seq |
| Software, algorithm | kallisto | *Bray et al., 2016* (doi:10.1038/nbt.3519) | | RNA-seq |
| Software, algorithm | Scaffold 4.9 | http://www.proteomesoftware.com/products/scaffold/ | | Mass-Spectrometry |
| Software, algorithm | Gorilla | *Eden et al., 2009* (doi:10.1186/1471-2105-10-48) | | GO term enrichment |
| Other | BioTek Synergy H1 plate-reader | BioTek Instruments | | Instrument: Plate-reader |
| Oher | C1000Touch plate reader | Bio-Rad | | Instrument: qPCR device |
| Other | BD FACSAria II P0287 (BSL2) | BD | | Instrument: Cell sorting |
| Other | BD Fortessa X-30 H0081 | BD | | Instrument: Flow cytometry |
| Other | Illumina HiSeq 2500 | Illumina | | Instrument: Sequencer |
| Other | Illumina NextSeq 550 | Illumina | | Instrument: Sequencer |
| Other | Orbitrap Fusion Tribrid MS-MS instrument | Thermo Scientific | | Instrument: Mass-Spectrometry |

## Yeast strains

We used 160 yeast strains, 12 of which were obtained from other laboratories, including six from the GFP collection, and 148 that were built for this study (complete list in *Supplementary file 2*). All strains are based on two distinct genetic backgrounds: BY4741 (BY), which is closely related to the commonly used laboratory strain S288c, and RM11-1a (RM), a haploid offspring of a wild isolate from a vineyard, which is closely related to European strains used in wine-making. Both strains carried auxotrophic markers, and RM had been engineered earlier to facilitate laboratory usage (BY: *his3Δone leu2Δ0 met15Δ0 ura3Δ0*; RM: *can1Δ::STE2pr-URA3 leu2Δ0 HIS3(S288C allele) ura3Δ ho:: HYG AMN1(BY allele)*; *Supplementary file 2*). Most strains were built using conventional yeast transformation (*Gietz and Schiestl, 2007*) and DNA integration based on homologous recombination. Integrated DNA fragments were produced by PCR (Phusion Hot Start Flex NEB M0535L, following manufacturer protocol, annealing temperature: 57°C, 36 cycles, final volume: 50 µl) and gel purified (Monarch DNA Gel Extraction Kit, NEB T1020L), with primers carrying 40 to 60 bp overhanging homologous sequence as required. All primers are available in *Supplementary file 3*. For transformation, fresh cells from colonies on agar plates were grown in YPD media (10 g/l yeast extract, 20 g/l peptone, 20 g/l glucose) overnight at 30°C. The next day, 1 ml of the culture was inoculated in an Erlenmeyer flask containing 50 ml of YPD and grown under shaking at 30°C for 3 hr to reach the late log phase. Cells were harvested by centrifugation and washed once with pure sterile water and twice with transformation buffer 1 (10 mM TrisHCl at pH8, 1 mM EDTA, 0.1 M lithium acetate). We resuspended the cells in 100 µl of transformation buffer 1, added 50 µg of denatured salmon sperm carrier DNA (Sigma #D7656) and 1 µg of the DNA fragment to be integrated, and incubated for 30 min at 30°C. Alternatively, when transforming a replicative plasmid, we used 0.1 µg of plasmid DNA and skipped this first incubation. We added 700 µl of transformation buffer 2 (10 mM TrisHCl at pH8, 1 mM EDTA, 0.1 M lithium acetate, 40% PEG 3350) and performed a second incubation for 30 min at 30°C. A heat shock was induced by incubating the cells at 42°C for 15 min. The transformed cells were then washed twice with sterile water. If the selective marker for the transformation was an antibiotic resistance gene, the cells were resuspended in 1 ml of YPD, allowed to recover for 2 hr at 30°C, and spread on a YPD plate (2% agar) containing the antibiotic (200 ng/l G418, 100 ng/l nourseothricin sulfate/CloNAT, or 300 ng/l hygromycin B). Alternatively, if the transformation was based on complementation of an auxotrophy, the cells were resuspended in 1 ml of sterile water and spread on a plate containing minimal media lacking the corresponding amino acid or nucleotide (YNB or Synthetic Complete (SC): 6.7 g/l yeast nitrogen base (VWR 97064–162), 20 g/l glucose, with or without 1.56 g/l SC -arginine -histidine -uracil -leucine (Sunrise science 1342–030), complemented as needed with amino acids: 50 mg/l histidine, 100 mg/l leucine, 200 mg/l uracil, 80 mg/l tryptophan). After two to three days of incubation at 30°C, colonies were streaked on a fresh plate containing the same selection media to purify clones arising from single, transformed cells. DNA integration in the correct location was confirmed by PCR (Taq DNA Polymerase NEB M0267L, following manufacturer protocol, annealing temperature: 50°C, 35 cycles, final volume: 25 µl, primers in *Supplementary file 3*). To store the constructed strains, we regrew the validated colony on a new selection media plate overnight at 30°C, scraped multiple colonies, resuspended the cells in 1.4 ml of YPD containing 20% glycerol in a 2 ml screw cap cryotube and froze them at −80°C.

## Plasmids

We constructed seven plasmids: three plasmids that do not replicate in yeast and that carry the GFP-gRNA tag, the CRISPR reporter, and $Z_3$EV system, respectively (*Figure 1—figure supplement 1*), and four yeast-replicating plasmids to investigate the quantitative properties of our reporter (*Figure 2—figure supplement 3*). These plasmids were constructed through multiple rounds of cloning using DNA fragments from yeast DNA or plasmids acquired from Addgene (kind gifts from John McCusker: Addgene #35121–22, from Michael Nick Boddy: Addgene #41030, from Benjamin Glick: Addgene #25444, from Timothy Lu: Addgene #64381, #64389, #49013; complete list of plasmids in *Supplementary file 4*). Plasmids were assembled using Gibson assembly (NEBuilder HiFi DNA Assembly Cloning Kit, NEB E5520S). Fragments were either PCR amplified with a least 15 bp overlap at each end (Phusion Hot Start Flex NEB M0535L, manufacturer protocol, annealing temperature: 57°C, 36 cycles, final volume: 50 µl, primers in *Supplementary file 3*) or obtained by restriction digestion of already existing plasmids (also shown in *Supplementary file 3*).

The fragment encoding the gRNA tag, containing the two ribozymes and the gRNA sequence itself, was purchased as a 212 bp double-stranded DNA oligo from IDT (we used the 'C3' gRNA from *Farzadfard et al., 2013*, as it was reported to provide the highest reporter gene expression). The synthetic polyA tail following the GFP sequence (*Figure 1A*) was introduced by using a PCR primer containing 45 thymines in its overhang sequence (primer OFA0038 in *Supplementary file 3*). Fragments for assembly were purified using agarose electrophoresis and gel extraction (Monarch DNA Gel Extraction Kit, NEB T1020L). For assembly, the given fragments were mixed at equi-molar amounts of 0.2–0.5 pM in 10 µl. Assembly was done by addition of 10 µl of NEBuilder HiFi DNA Assembly Master Mix and incubation at 50°C for 60 min. From this reaction, 2 µl of the final products were transformed into *E. coli* competent cells (10-beta Competent E.coli, NEB C3019I) through an incubation of 30 min on ice and a heat shock of 30 s at 42°C. Transformed cells were spread on LB plates (10 g/l peptone, 5 g/l yeast extract, 10 g/l sodium chloride, 2% agar) containing 100 mg/l ampicillin and grown overnight at 37°C. After cloning, the final plasmids were extracted (Plasmid Miniprep Kit, Zymo Research D4036) and verified by restriction enzyme digestion or PCR (Taq DNA Polymerase NEB M0267L, 30 cycles, 25 µl final volume, primers in *Supplementary file 3*). We also verified by Sanger sequencing that the gRNA tag in the plasmid had no mutation. To store the plasmids, the host bacteria were grown in LB with ampicillin overnight at 37°C and 1 ml of the culture was mixed with 0.4 ml of a sterile solution containing 60% water and 40% glycerol. The cells were stored at −80°C. The three plasmids containing the different parts of the reporter are available on Addgene (ID #157656, #157658, and #157659) along with their full DNA sequence.

## Plate reader-based fluorescence measurements

Yeast fluorescence was measured in 24 hr time courses during overnight growth in a BioTek Synergy H1 plate-reader (BioTek Instruments). Fresh cells from agar plates were inoculated in 100 µl of minimal YNB media (*Figure 1*) or SC media (all other figures) containing any complements necessary for growth of auxotrophic strains, at an initial optical density at wavelength 600 nm ($OD_{600}$) of 0.05 in a 96-well flat bottom plate (Costar #3370). We conducted the experiments in *Figure 1* in the presence of G418 (which requires the use of YNB + glutamate) because, at the time, we were concerned about potential loss of the reporter construct from the genome due to recombination between its flanking sequences. This concern later turned out to be unfounded. The plates were sealed with a Breathe Easy membrane (Diversified Biotech). Cells were grown in the plate reader at 30°C and with circular agitation in between fluorescence acquisition. During each acquisition, performed every 15 min, we recorded $OD_{600}$, GFP fluorescence (read from bottom, excitation 488 nm, emission 520 nm, 10 consecutive reads averaged, gain set to 'extended') and mCherry fluorescence (read from bottom, excitation 502 nm, emission 532 nm, 50 consecutive reads averaged, gain set to a value of 150). We took 97 measurements during 24 hr of growth, unless individual runs were manually terminated early.

Raw measurements of $OD_{600}$ and fluorescence were processed using R version 3.5.1 (https://www.r-project.org/, scripts and raw data available at the github repository at *Brion, 2020*; https://github.com/BrionChristian/Simultaneous_RNA_protein_QTLs; copy archived at swh:1:rev:74628305678f30d5b126745e5c6cd6cec4091c12). 'Blank' values from wells with no cells were subtracted from $OD_{600}$ and fluorescence measurements of wells that had been inoculated with cells. $OD_{600}$ was log-transformed and manually inspected to identify the late log phase, i.e. a time point about 3/4 into the exponential growth phase. This stage was identified separately for each well, and usually corresponded to an $OD_{600}$ of 0.1–0.3. We extracted the $OD_{600}$ and fluorescence measurements at the five time points centered on our selected time point. The mCherry and GFP fluorescence ratios were calculated as the ratio between the fluorescence and the $OD_{600}$ at these five time points (example in *Figure 1B–CB*), allowing us to estimate fluorescence while correcting for culture density. Focusing on the late log phase allowed measurements at higher cell density to provide more robust fluorescence reads. Growth rates were estimated as the slope of a linear fit of the log of $OD_{600}$ over time.

## RNA quantification by qPCR

### Cell harvest

We quantified mRNA and gRNA abundance by quantitative real-time reverse-transcription PCR of RNA extracted from exponentially growing cells. Cells were grown in either 50 ml of medium (YNB with auxotrophic complements, results shown in *Figure 1D*) in shaking Erlenmeyer flasks or in 1.2 ml of media (SC with auxotrophic complements and estradiol (Sigma, E1024-1G), *Figure 2C–EC* and *Figure 6F*) in a shaking 2 ml 96-deep-well plate. The $OD_{600}$ was monitored to identify the second half of the exponential growth phase (corresponding to an $OD_{600}$ of 0.35–0.45 $OD_{600}$ in flasks, and 0.20–0.30 in the deep-well plates). At this point, GFP and mCherry fluorescence ratios were recorded in a BioTek Synergy H1 plate reader (BioTek Instruments). Cells were then harvested immediately. Cells were washed with sterile water through either short centrifugation using 5 ml of culture from flasks, or vacuum-filtration through a 96-well filter plate (Analytical Sales 96110–10) using the entire remaining 1 ml of culture from the deep-well plate. Cells were then immediately flash-frozen in either isopropanol at −80°C (pellet from flask) or liquid nitrogen (filter plate) and stored at −80°C until RNA extraction.

### RNA extraction from flasks

To extract the RNA from cells grown in flasks, we used the ZR Quick-RNA Kit (Zymo Research R1054). Frozen cell pellets were resuspended in 800 µl RNA Lysis Buffer one from the kit and transferred to a ZR BashingBead Lysis Tube. The cells were shaken in a mini-bead beater (BioSpec Products) for ten cycles of one minute in the beater, one minute on ice. Cell debris and beads were centrifuged for one minute at full speed and 400 µl of supernatants were transferred into Zymo-Spin IIIC Columns. The columns were centrifuged for one minute, and 400 µl 100% ethanol was added to the flow-through. After mixing, the flow-throughs were transferred into Zymo-Spin IIC Columns and centrifuged for 1 min to bind the RNA and DNA to the columns. The columns were washed with 400 µl RNA Wash Buffer from the kit. DNA was digested in columns by adding a mixture of 5 µl DNase I and 75 µl DNA Digestion Buffer from DNase I Set kit (Zymo Research E1010) followed by a 15 min incubation at room temperature. The columns were then washed three times with 400 µl RNA Prep Buffer, 700 µl RNA Wash Buffer, and 400 µl RNA Wash Buffer. RNA was eluted in 50 µl DNase/RNase-free water, quantified using Qubit RNA BR or HS Assay Kit (Thermo Fisher Scientific Q10210 or Q52852), and stored at −20°C.

### RNA extraction from 96-well plates

To extract the RNA from cells grown in 96-well plates, we used the ZR RNA in-plate extraction kit (ZR-96 Quick-RNA Kit, Zymo Research R1052), which followed the same protocol as the flask RNA extraction above, with a few minor differences. Bead-beating was done in an Axygen 1.1 ml plate (P-DW-11-C-S) with 250 µl of acid washed 425–600 µm beads (Sigma G8722) per well, sealed with an Axymat rubber plate seal (AM-2ML-RD-S). RNA purified from 200 µl of the resulting supernatant. DNA digestion and washing steps were done on Silicon-A 96-well plates from the kit. The RNA was eluted in 30 µl of DNase/RNase-free water, quantified, and stored at −20°C.

### Reverse-transcription and qPCR

RNA was reverse-transcribed using the GoScript RT kit (Promega A5000) following the kit protocol. We performed negative controls, no-enzyme and no-primer, which generated no qPCR signals. Quantitative PCRs were done in a 96-well plate (Bio-Rad HSP9645) using GoTaq qPCR kit (Promega A6001). Plates were sealed using a microseal 'B' Adhesive Seal (Bio-Rad MSB1001) and the reaction progress was recorded during 40 cycles using a C1000Touch plate reader (Bio-Rad). We quantified four different parts of the tag cDNA (GFP, Hh ribozyme cleavage, gRNA, and HDV ribozyme cleavage, *Figure 1D*), as well as *ACT1* cDNA as a reference gene. Primer sequences are in *Supplementary file 3*. The primers were tested and calibrated by running qPCR measurements on nuclear DNA extracts at a range of known input concentrations (*Figure 1—figure supplement 2*).

## Segregant populations

BY strains (BY4741 background) carrying a given GFP-gRNA-tag and RM (YFA0198) carrying the CRISPR reporter and the SGA marker were mixed for crossing on a plate with medium that allows

only hybrids to grow (SC agar -leucine -histidine). Growing cells were streaked on the same medium, and a single hybrid colony was kept for storage and for generating the segregant population. For sporulation, hybrid strains were incubated in sporulation medium (2.5 g/l yeast extract, 2.5 g/l glucose, 15 g/l potassium acetate, 200 mg/l uracil, 100 mg/l methionine) at room temperature under vertical rotation in a glass tube for seven days. After verifying sporulation under a light microscope, 1 ml of medium containing the tetrads was pelleted (13,000 rpm for 5 min) and resuspended in 300 µl of sterile water containing about 15 µg of zymolyase. The resulting ascii were digested at 30°C for 30 min with agitation. Spores were separated by vortexing for about 15 s, and 700 µl of pure sterile water was added to the tube. We spread 250 µl of this spore suspension on a plate containing segregant selection media (SC agar, 50 mg/l canavanine, -uracil -leucine) allowing growth of haploid segregants carrying the following three alleles: (1) cells with mating type MATa, selected via the SGA marker with URA3 under control of the STE2 promoter, which resulted in a ura+ phenotype only in MATa cells, (2) the SGA marker integrated at the CAN1 gene (whose deletion conferred canavanine resistance), which also selected for the CRISPR reporter that we had integrated at NPR2, the gene next to CAN1, (3) the given gene of interest tagged with the GFP-gRNA tag and LEU2 selectable marker. We used SC because this medium complemented a growth defect of strains with an NPR2 deletion in YNB medium. After three days of incubation at 30°C, segregants were harvested by scraping the entire plate in 10 ml of sterile water. Cells were centrifuged, resuspended in 3 ml of segregant selection media, and incubated at 30°C for 1.5 hr. To store these genetically diverse segregant populations, 1 ml of the culture was mixed with 0.4 ml of a sterile solution containing 60% water and 40% glycerol in a 2 ml screw cap cryo tube and frozen at −80°C.

## Cell sorting for QTL mapping

One day before cell sorting, the segregant population was thawed from the −80°C stock, mixed well, and 8 µl of culture were used to inoculate 5 ml of segregant selection media. The cells were reactivated with an overnight growth at 30°C under shaking. The next day, 1 ml of the growing culture was transferred to a new tube containing 4 ml of segregant selection medium (see above) and grown for an additional two hours before cell sorting, roughly corresponding to the middle of the exponential growth phase.

Cell sorting was performed on a BD FACSAria II P0287 (BSL2) instrument at the University of Minnesota Flow Cytometry Resource (UFCR), without compensation. Cells were gated to exclude doublet and cellular fragments. To focus on cells in approximately the same stage of the cell cycle, an additional gate selected cells in a narrow range of cell size as gauged by the area of the forward scatter signal (FSC). From the cells within this gate, we sorted five populations per experiment, each comprising 10,000 cells: (1) a control population from the entire gate without fluorescence selection, (2) the 3% of cells with the lowest GFP fluorescence, (3) the 3% of cells with the highest GFP fluorescence, (4) the 3% of cells with the lowest mCherry fluorescence, and (5) the 3% of cells with the highest mCherry fluorescence. Each population was collected into 1 ml of segregant selection medium. After overnight growth at 30°C, 0.9 ml of culture was mixed with 0.4 ml of a sterile solution containing 60% water and 40% glycerol, and frozen at −80°C until sequencing. The remaining 0.1 ml were inoculated into 0.9 ml of segregant selection medium and grown for 3 hr before analyzing the population using flow cytometry (see below). In total, we obtained 125 sorted populations from 25 experiments across the ten tagged genes, with 1 to 6 biological replicates per gene, as well as the untagged population (*Figure 5—source data 1*). Sorting was done in four batches on different dates. Biological replicates were performed as independent sporulations of the stored diploid hybrids, and thus represent independent populations sorted in separate experiments.

## Flow cytometry

Single-cell fluorescence analysis was performed using cultures in the late log growth phase. We used a BD Fortessa X-30 H0081 flow cytometer at UFCR equipped with blue and yellow lasers and 505LP and 595LP filters to measure green (GFP) and red (mCherry) fluorescence, respectively. We did not use compensation. Forward scatter (FSC), side scatter (SSC), GFP, and mCherry fluorescence were recorded for 50,000 cells, excluding doublets and cellular debris. The voltaic gains were set as follows: 490 for FSC, 280 for SSC, 500 for GFP, and 600 for mCherry. We monitored for possible cross-contamination from cells retained in the instrument using strains expressing either only GFP or

mCherry, and observed no cross-contamination. Recorded data on. fsc files were analysed using R and the flowCore package (*Hahne et al., 2009*). Raw data and scripts are accessible on github (https://github.com/BrionChristian/Simultaneous_RNA_protein_QTLs). The data were filtered to discard outlier cells based on unusual FSC and SSC signals. We used the fluorescence data from the sorted populations to determine correlations between red and green fluorescence, as well as heritability (*Figure 3—figure supplement 1* & *Figure 3—figure supplement 2*). For these analyses, fluorescence values were corrected for cell size (FSC) by calculating the residuals of a loess regression of fluorescence on FSC. Loess regression avoided the need to assume a specific mathematical relationship between the two parameters (*Figure 3—figure supplement 3*).

## DNA extraction and sequencing

DNA extraction for whole-genome sequencing was performed in 96-well plate format using E-Z-96 Tissue DNA kits (Omega D1196-01). The stored, sorted populations were thawed, mixed, and 450 µl transferred into a 2 ml 96-deep-well plate containing 1 ml of segregant selection medium for an overnight growth at 30°C. The plate was centrifuged for 5 min at 3700 rpm, and the supernatant was removed by quick inversion of the plate. Then, 800 µl of Buffer Y1 (182 g/l sorbitol, 0.5 M EDTA, pH 8, 14.3 mM β-mercaptoethanol, 50 mg/l zymolyase 100T) were added to the pellets, and the cells were resuspended and incubated for 2 hr at 37°C. The spheroplasts were centrifuged and the supernatant discarded. The pellets were resuspended in 200 µl of TL buffer and 25 µl of OB Protease Solution from the kit and incubated overnight at 56°C. The next day, RNA was denatured by addition of 5 µl of RNAse A (20 mg/ml) and incubated at room temperature for 5 min. After addition of 450 µl of BL Buffer from the kit, the mixture was transferred onto a E-Z 96 column DNA plate and centrifuged at 3700 rpm for 3 min. The columns were washed once with 500 µl of HBC Buffer and three times with 600 µl of DNA Wash Buffer from the kit. After an additional centrifugation to dry the column, the DNA was eluted in 100 µl of pure sterile water, quantified using Qubit dsDNA HS Assay Kit (Thermo Fisher Scientific Q32854) and stored at 4°C for library preparation the next day.

Library preparation for Illumina sequencing was performed using Nextera DNA Library Prep kit (Illumina) with modifications. The tagmentation was done on 5 ng of DNA using 4 µl of Tagment DNA buffer ('TD' in the kit) and 0.25 µl of Tagment DNA enzyme (corresponding to a 20-fold dilution of 'TDE1' from the kit) and incubating for 10 min at 55°C. Fragments were amplified with index primers (8 Nextera primers i5 and 12 Nextera primers i7, for up to 96 possible multiplex combinations) on 10 µl tagmented DNA by adding 1 µl of each primer solution (10 µM), 5 µl of 10X ExTaq buffer and 0.375 µl of ExTaq polymerase (Takara) and water to a final volume of 50 µl. The amplification was run for 17 PCR cycles (95°C denaturation, 62°C annealing, 72°C elongation). 10 µl of each reaction were pooled for multiplexing and run on a 2% agarose gel. DNA that migrated between the 400 and 600 bp was extracted using Monarch DNA Gel Extraction Kit (NEB T1020L). The pooled library DNA concentration was determined using Qubit dsDNA BR Assay Kit (Thermo Fisher Scientific Q32853), and submitted for sequencing. Sequencing was performed at the University of Minnesota Genomics Core (UMGC). Our 125 populations were processed in four batches extracted and sequenced at different times. Two were sequenced using an Illumina HiSeq 2500 (high-output mode; 50 bp paired-end) and two were sequenced using an Illumina NextSeq 500 (mid-output mode, 75 bp paired-end). Read coverage ranged from 5-fold to 24-fold coverage of the genome (median: 13-fold). The reads are available on NCBI SRA via BioProject PRJNA644804.

## QTL mapping

For each sorted and sequenced population, reads were filtered (MAPQ $\geq$30) and aligned to the *S. cerevisiae* reference genome (version sacCer3, corresponding to BY strain) using BWA (*Li and Durbin, 2009*, command: *mem -t*). We used samtools (*Li et al., 2009*, command: *view -q 30*) to generate bam files and collapse PCR duplicates using the *rmdup* command. We used 18,871 variants previously identified as polymorphic and reliable between RM and BY (*Bloom et al., 2013*; *Ehrenreich et al., 2010*) (list available on github: https://github.com/BrionChristian/Simultaneous_RNA_protein_QTLs, samtools: *mpileup -vu -t INFO/AD -l*), generating vcf files with coverage and allelic read counts at each position for each population.

The vcf files were processed in R to identify bulk segregant analysis QTLs using code adapted from *Albert et al., 2014b* (available on github: https://github.com/BrionChristian/Simultaneous_

RNA_protein_QTLs). Briefly, for plotting the results, the allele frequency of the reference (that is, BY) allele was calculated at each position in each population. Random counting noise was smoothed using loess regression, and the allele frequency of a given 'low' fluorescence population subtracted from its matched 'high' fluorescence population to generate ΔAF. A deflection from zero indicated the presence of a QTL. To identify significant QTLs, we used an R script that implemented the MUL-TIPOOL algorithm (*Edwards and Gifford, 2012*), which calculates LOD score based on ΔAF and depth of read coverage in bins along the genome. We used MULTIPOOL output to call QTLs as peaks exceeding a given significance threshold (see below), along with confidence intervals for the peak location corresponding to a 2-LOD drop from the peak LOD value. We applied the MULTI-POOL algorithm using the following parameters: bp per centiMorgan: 2,200; bin size: 100 bp, effective pool size: 1000. We excluded variants with extreme allele frequencies of >0.9 or <0.1. We initially set a permissive detection threshold of LOD >3.0 to identify a set of candidate QTLs, which we then integrated across replicates (507 QTLs, *Figure 4—source data 1*). A second, more stringent, threshold of LOD >4.5 was then applied to retain only significant QTLs based on our estimated false discovery rate (FDR).

To estimate FDR, we applied the multipool QTL detection algorithm to pairs of populations sorted into the same gates in different replicates. Any 'QTLs' in such comparisons must be due to technical or biological noise. We restricted these analyses to replicates sequenced on the same instrument, resulting in 80 inter-replicate comparisons. From these data, we calculated the FDR as a function of the significance threshold (*thr*): $FDR_{thr} = (NrepQTL_{thr}/Nrep) / (NfluoQTL_{thr}/Nfluo)$, where $NrepQTL_{thr}$ is the number of false 'QTLs' from comparing the same gate across replicate populations at a LOD score threshold of *thr*, *Nrep* is the number of such inter-replicate comparisons (*Nrep* = 80), $NfluoQTL_{thr}$ is the number of fluorescence-QTLs at a LOD threshold of *thr*, and *Nfluo* is the number of high vs. low fluorescence comparisons (*Nfluo* = 48; the untagged experiment was excluded). At a significance threshold of LOD = 4.5, the estimated FDR was 7.3% (*Figure 4—figure supplement 1*), which we used to call significant QTLs. For some overlap analyses (see below), we used a threshold of LOD = 3.0, which corresponded to an FDR of 13%.

To call significant QTLs across replicates, we first scanned each replicate for QTLs at a permissive threshold of LOD > 3.0. Second, at each resulting QTL peak position, we averaged ΔAF and LOD scores across all available replicates without applying a LOD filter to each replicate. Third, we collapsed groups of overlapping QTLs, which we defined as QTLs whose peaks were within 75,000 bp of each other in the different replicates. For each group of these overlapping QTLs, we averaged the LOD scores, the ΔAFs, the peak positions, and the location confidence intervals to form one merged QTL. Of the resulting merged QTLs, we retained those that exceeded our stringent significance threshold of LOD ≥ 4.5.

To gauge reproducibility of these significant QTLs, we counted the number of replicates in which a given QTL had been detected at the permissive LOD >3.0, using the same definition of positional overlap as above. The majority (74%) of significant QTLs were shared across all the corresponding replicates. Two tagged genes had more than two replicates (*GPD1* and *UGP1*). For these genes, requiring *all* replicates to be significant is conservative. Therefore, we also estimated the average reproducibility of all mRNA-QTLs or all protein-QTLs by calculating the average fraction of replicates that had a QTL at a given merged QTL:

$$fraction\_overlap = mean[(NshareQTL_{ij} - 1)/(Nrep_j - 1)]$$

Here, $NshareQTL_{ij}$ is the number of replicates for which the QTL *i* is detected for the tagged gene *j* at LOD > 3, and $Nrep_j$ is the number of replicates performed for the tagged gene *j*. Note that if only a single replicate has a QTL at a given merged QTL, this fraction takes on a value of zero because in such a case, there is no overlap among replicates at this QTL. The observed *fraction_overlap* was 0.76 for the protein-QTLs and 0.78 for the mRNA-QTLs.

As a measure of effect size for a QTL, we used ΔAF. Briefly, in the presence of a QTL, there are two subpopulations of segregants that carry the two QTL alleles. The effect of the QTL is given by the difference in mean between these two subpopulations, assuming identical standard deviations for the two alleles. During FACS of the combined population with both alleles, cells that exceed the threshold for inclusion in the high population are enriched for the allele that increases expression, and vice versa for the low population. The ratio of cells that carry the two alleles in each extreme

population determines the allele frequency at the locus in that population, and subtracting these allele frequencies for the two populations gives the ΔAF metric. For QTLs with larger effect sizes, the two subpopulations are further apart in phenotype. This increases the degree of genotype enrichment in the two extreme populations, as reflected in a larger ΔAF.

## Comparison of mRNA-QTLs and protein-QTLs

To compare mRNA-QTLs and protein-QTLs of the same gene, we first considered all merged QTLs that exceeded a permissive threshold of LOD > 3.0 (after merging replicates as described above). We considered an mRNA-QTL and a protein-QTL for the same gene with overlapping confidence intervals as a QTL pair across mRNA and protein. We manually curated the result of this overlap analysis for six cases; after curation, QTLs on chromosomes XV (*ARO8*), VIII (*MTD1*), XIII (*CYC1*) and XIII (*RPS10A*) were considered to be pairs, and QTLs on chromosome V (*GPD1*) and XIV (*MTD1*) were considered to be mRNA-specific.

From this initial set, we retained those QTL pairs at which the given mRNA and/or protein QTL met a more stringent LOD score of >4.5 (FDR = 7.2%). Applying this higher threshold only after the more permissive overlap analysis allowed us to consider QTL pairs even if one of the paired QTLs did not pass the strong significance threshold of LOD >4.5. As an example, we considered overlapping QTLs on chromosome XI that affected *OLE1* expression (mRNA-QTL LOD = 4.4, protein-QTL LOD = 15.5) to be a pair even if the mRNA LOD score was below the stringent significance threshold. In such cases, we deemed it more conservative to assume that the weaker QTL exists but narrowly failed to reach significance than to declare the stronger QTL as specific for mRNA or protein. We discarded all QTLs located between 350 and 550 kb on chromosome XIV, as this region may affect mCherry fluorescence independently of the tagged gene.

We distinguished four types of QTLs (*Figure 5A* and *Figure 5—figure supplement 1*). The shared QTL pairs either had similar effects on mRNA and protein abundance (16 QTL pairs, defined as having the same sign of ΔAF), or discordant effects on mRNA and protein (7 QTL pairs, different sign of ΔAF). All QTLs that were not part of a pair were considered to be specific (11 mRNA-specific QTLs, 52 protein-specific QTLs).

Finally, we conducted an analysis of mRNA or protein QTL effect sizes and directions that avoided having to define potentially paired QTLs as significant or not. For each mRNA-QTL (or protein-QTL), we extracted the ΔAF from the protein-QTL (or mRNA-QTL) data at the same exact position, irrespective of significance in the other data. We used these values to compute correlations of effects and to examine shared directionality of effects between mRNA-QTLs and protein-QTLs (*Figure 5B*, *Figure 5—source data 2*).

## Allelic engineering for *YAK1* fine-mapping

To obtain strains with scarless allelic swaps in haploids, we used a strategy based on double-cut CRISPR-swap (*Lutz et al., 2019*). We flanked each of the four tiles to be switched by two resistance markers (*hphMX* and *KanMX*) using our regular yeast transformation protocol (see above). The yeast were then transformed with 100 ng of CRISPR-Swap plasmid (pFA0055-gCASS5a, Addgene plasmid # 131774) and 1 μg of DNA repair template amplified either from BY, BY *GPD1-GFP*, or RM. The transformed cells were spread on SC -leucine plates, selecting for the presence of the plasmid expressing *CAS9* and a gRNA targeting and cleaving a sequence present in both of the two resistance cassettes. We used strains in which *GPD1* was tagged with *GFP* but not the gRNA tag, as the gRNA in our tag would likely have directed *CAS9* to cleave the mCherry promoter. Cleavage of both cassettes resulted in the region in between the resistance cassettes to be replaced by the repair template. Transformed clones were screened for the double loss of antibiotic resistance to identify those with successful editing.

We introduced the 148,659 G→A variant, which we had detected through sequence analysis (see below), in *YAK1* by single-cut CRISPR-swap (*Lutz et al., 2019*). We replaced the *YAK1* sequence with a *hphMX* resistance cassette insertion to create *yak1Δ::hphMX*. We then delivered the CRISPR-Swap plasmid along with a repair template DNA produced by fusion PCR to carry either the G or A allele at the variant position (primers OFA0874 to OFA0881 in *Supplementary file 3*). Five clones of each allele, (*YAK1*^wt and *YAK1*^Q578*) were confirmed by Sanger sequencing (primers OFA0883 and OFA0882 in *Supplementary file 3*).

## Sequence analyses to identify the *YAK1* and other new DNA variants

To search for new variants in our populations that were not known to be present in the BY and RM strains, we used the sequencing reads from each selected segregant population. In each of 125 populations, we analyzed bam files after collapsing PCR duplicates. We applied samtools *mpileup* (*–minBQ 0*) and bcftools *call* (*-vc*), either locally in the region affecting Gpd1-GFP (*-r chrX:146000–150000*) or on the whole genome to generate vcf files containing variant information. The vcf files were merged in R to generate matrices of polymorphic positions along with their allele frequency and coverage. The allele frequencies of the *YAK1* polymorphisms were plotted along the genome for each population (*Figure 6—figure supplement 1*). We excluded all 47,754 previously known BY/RM variants from the whole-genome polymorphism matrices, and also removed variants with a bcftools quality score below 30. Among the 7624 remaining variants sites, 5822 were identified in only one or two populations and were deemed to be sequencing errors. Only one variant was shared across most (71 out of 75) of the populations created from strains from the GFP collection (strains tagged at *ARO8*, *BMH2*, *GPD1*, *MTD1*, and *UGP1*; *Figure 5—source data 1*) and absent in all other populations. This variant was the 148,659 G-to-A SNV in *YAK1*.

## *YAK1* genotyping

The region containing the $YAK1^{Q578*}$ variant was PCR-amplified from genomic DNA isolated from strains carrying *FAA4*-GFP and YMR315W-GFP (GFP collection), *PGM1*-TAP, *NOT5*-TAP, *EMI2*-TAP, *TUB1*-TAP (TAP-tag collection), BY4741 (ATCC 201388), YLK1879 (a BY strain from the Kruglyak lab) and YLK1950 (an RM strain from the Kruglyak lab) using primers OFA0883 and OFA0882 (*Supplementary file 3*). The resulting PCR product was Sanger sequenced using OFA0883 to genotype the *YAK1* variant. The $YAK1^{Q578*}$ variant was observed only in the strains obtained from the GFP and TAP-tag collections (*Ghaemmaghami et al., 2003*; *Huh et al., 2003*).

## Differential expression analysis by RNA-seq and mass spectrometry

### Cell harvest

We quantified RNA and protein from five biological replicates (different clones obtained after CRISPR-Swap) of $YAK1^{wt}$ and $YAK1^{Q578*}$. For each of these 10 strains, fresh colonies were used to inoculate 5 ml of SC medium (completed with uracil, arginine, histidine, and leucine) and the culture was grown overnight at 30°C. The next day, 50 ml of SC media in an Erlenmeyer flask was inoculated with the overnight culture to an initial $OD_{600}$ of 0.05 and were grown under shaking at 30°C. When the $OD_{600}$ reached 0.35–0.45 (late log phase), four aliquots of 7 ml of the culture was transferred to a falcon tube, centrifuged, and washed with 1 ml of sterile PBS buffer at 30°C (Phosphate Buffered Saline, pH 7.5). The pellets were immediately frozen in liquid nitrogen. For each strain, one cell pellet was used for RNA-seq and another for protein mass spectrometry.

### RNA extraction and sequencing

RNA extraction and library preparation were conducted as described in *Lutz et al., 2019*. Briefly, RNA extraction was done in two batches that each contained equal numbers of clones from the two groups (first batch: clones 1 and 2 of BY $YAK1^{wt}$ and clones 1 and 2 of BY $YAK1^{Q578*}$, second batch: clones 3, 4, and 5 of BY $YAK1^{wt}$ and clones 3, 4, and 5 of BY $YAK1^{Q578*}$). We used ZR Fungal/Bacterial RNA mini-prep kit with DNase I digestion (Zymo Research R2014), following the kit manual. RNA was eluted in 50 µl of RNase/DNase free water, quantified, and checked for integrity on an Agilent 2200 TapeStation. All RNA Integrity Numbers were higher than 9.5 and all concentrations were above 120 ng/µl. The RNA samples were stored at −20°C until use. Poly-A RNA selection was done using 550 ng of total RNA using NEBNext Poly(A) mRNA Magnetic Isolation Module (NEB E7490L), processing all samples in one batch. We prepared the library using NEB Ultra II Directional RNA library kit for Illumina (NEB E7760) used dual index primers (from NEBNext Multiplex Oligos for Illumina, NEB E7600S) for multiplexing, and amplified the library for ten cycles. The libraries were quantified using Qubit dsDNA HS Assay Kit (Thermo Fisher Scientific Q32854) and pooled at equal mass for sequencing using 75 bp single-end reads on Illumina NextSeq 550 at UMGC. The sequencing reads are available on NCBI GEO via BioProject PRJNA644804. Using the trimmomatic software (*Bolger et al., 2014*), reads were trimmed of adapters and low quality bases and filtered to be at least 36 bp long. Reads were then pseudo-aligned to the *S. cerevisiae* transcriptome (Ensembl build

93) and counted using kallisto (*Bray et al., 2016*). We used RSeQC to calculate Transcript Integrity Numbers (TIN) which provided an estimation of alignment quality for each gene of each sample. We excluded any gene with at least one read count of zero or at least one TIN of zero across the ten samples. After filtering, 5755 genes remained for analysis. Differential expression analysis was performed using the DESeq2 R package (*Love et al., 2014*), using the extraction batch information as covariate. DESeq2 provided, for each gene, the $\log_2$-fold-change ($YAK1^{Q578*}$ vs. $YAK1^{wt}$) and the p-value adjusted for multiple-testing using the Benjamini-Hochberg method (*Benjamini and Hochberg, 1995*; *Figure 7—source data 1*).

## Protein extraction and mass spectrometry

Protein extraction and quantification using mass spectrometry was performed by the Center for Mass Spectrometry and Proteomics at the University of Minnesota. Briefly, cells from the pellets were lysed by sonication (30%, 7 s in Branson Digital Sonifier 250) in protein extraction buffer (7 M urea, 2 M thiourea, 0.4 M triethylammonium bicarbonate pH 8.5, 20% acetonitrile and 4 mM tris(2-carboxyethyl)phosphine). The proteins were extracted using pressure cycling (Barocycler Pressure Biosciences NEP2320, 60 cycles of 20 s at 20 kpsi and 10 s at 0 kpsi), purified by centrifugation in 8 mM iodoacetamide (10 min at 12000 rpm), and quantify by Bradford assay. For each sample, 40 μg of extracted proteins were fragmented by trypsin digestion (Promega Sequencing Grade Modified Trypsin V5111), and labeled by tandem mass tag isotopes (TMT10plex Isobaric Label Reagent Set, Thermo Scientific 90110). All the tagged samples were pooled. Peptides were separated first based on hydrophobicity through two consecutive liquid chromatographies, followed by separation based on mass per charge after ionization in the first mass spectrometry step. The second mass spectrometry step after high-energy collision-induced dissociation allowed for the identification of the peptide and the quantification of the TMTs. MS-MS analysis was conducted on an Orbitrap Fusion Tribrid instrument (Thermo Scientific). Database searches were performed using Proteome Discoverer software, and post-processing and differential expression analysis was done using Scaffold 4.9. Differential expression statistics per protein were computed on mean peptide abundances after inter-sample normalization. Normalized abundances for each of the 2590 detected proteins are provided in *Figure 7—source data 3*. We adjusted p-values provided by Scaffold using the Benjamini-Hochberg method.

## Protein and mRNA data comparison

To compare the effect of the *YAK1* variant on mRNA and protein abundance, we examined the $\log_2$ fold-change of the mRNA abundance (from DESeq2) and protein abundance (from Scaffold) for the 2577 genes present in both datasets. We considered genes that were differentially expressed in mRNA or protein (adjusted p-value<0.05) and that showed no evidence of difference in the other quantity (raw p-value>0.05), to be specifically affected at the mRNA or protein level (*Figure 7A*, *Figure 7—source data 4*). In various categories of differentially expressed genes, we looked for gene ontology enrichment using GOrilla (*Eden et al., 2009*) with the list of 2577 genes detected in both mass spectrometry and RNA-seq as the background set (*Figure 7—source data 2*).

# Acknowledgements

We thank Scott McIsaac and Leonid Kruglyak for yeast strains, Joshua Bloom for implementing the MULTIPOOL algorithm in R, and Mahlon Collins for the mCherry/GFP protein fusion flow cytometry data. We thank the University of Minnesota Genomics Center, the University of Minnesota's Center for Mass Spectrometry and Proteomics, and the University of Minnesota's Flow Cytometry Resource. We thank members of the Albert lab for critical feedback on the manuscript.

# Additional information

## Funding

| Funder | Grant reference number | Author |
| --- | --- | --- |
| National Institute of General | R35-GM124676 | Frank Wolfgang Albert |

Medical Sciences

| Alfred P. Sloan Foundation | FG-2018- 10408 | Frank Wolfgang Albert |

The funders had no role in study design, data collection and interpretation, or the decision to submit the work for publication.

### Author contributions

Christian Brion, Conceptualization, Data curation, Software, Formal analysis, Validation, Investigation, Visualization, Methodology, Writing - original draft, Writing - review and editing; Sheila M Lutz, Conceptualization, Validation, Methodology, Writing - review and editing; Frank Wolfgang Albert, Conceptualization, Software, Supervision, Funding acquisition, Methodology, Writing - original draft, Project administration, Writing - review and editing

### Author ORCIDs

Christian Brion (iD) https://orcid.org/0000-0002-3548-1401
Sheila M Lutz (iD) https://orcid.org/0000-0002-6729-4598
Frank Wolfgang Albert (iD) https://orcid.org/0000-0002-1380-8063

### Decision letter and Author response

Decision letter https://doi.org/10.7554/eLife.60645.sa1
Author response https://doi.org/10.7554/eLife.60645.sa2

## Additional files

### Supplementary files

• Supplementary file 1. Gene expression levels based on previous work: mRNA from RNA-seq *Albert et al., 2018* and protein from GFP fluorescence from *Huh et al., 2003*.

• Supplementary file 2. List of yeast strains.

• Supplementary file 3. List of PCR primers and DNA fragments.

• Supplementary file 4. List of plasmids.

• Transparent reporting form

### Data availability

Raw DNA reads from bulk segregant mapping are available via the NCBI BioProject PRJNA644804. Transcriptome sequencing data is available at GEO under accession GSE155998. Source Data files are available for Figures 4, 5, and 7.

The following datasets were generated:

| Author(s) | Year | Dataset title | Dataset URL | Database and Identifier |
|---|---|---|---|---|
| Brion C, Lutz S, Albert FW | 2020 | Differential RNA abundance between *S. cerevisiae* yeast BY4741 strains with YAK1_WT and YAK1_ Q578* | https://www.ncbi.nlm. nih.gov/geo/query/acc. cgi?acc=GSE155998 | NCBI Gene Expression Omnibus, GSE155998 |
| Brion C, Lutz S, Albert FW | 2020 | Simultaneous mRNA and protein quantification in single cells reveals post-transcriptional effects of genetic variation | https://www.ncbi.nlm. nih.gov/bioproject/ PRJNA644804/ | NCBI BioProject, PRJNA644804 |

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
