## [Decision Letter]

**Acceptance summary:**

We are excited to publish this work because it presents a novel, creative strategy for studying variation in RNA and protein expression in parallel. It then uses this strategy to simultaneously quantify RNA and protein expression in genetically variable cells and examines the genetic basis for this expression variation. Consistent with prior work, they find different loci contributing to variation in RNA and protein expression, but this work greatly strengthens this observation because the new methodology removes many other possible sources of differences between the two.

**Decision letter after peer review:**

Thank you for submitting your article "Simultaneous quantification of mRNA and protein in single cells reveals post-transcriptional effects of genetic variation" for consideration by *eLife*. Your article has been reviewed by three peer reviewers, and the evaluation has been overseen by Patricia Wittkopp as the Senior and Reviewing Editor. The following individual involved in review of your submission has agreed to reveal their identity: David Gresham (Reviewer #1).

The reviewers have discussed the reviews with one another and the Reviewing Editor has drafted this decision to help you prepare a revised submission.

This paper presents a creative, novel strategy for quantifying expression of RNA and protein of the same genes in the same cells. While some concerns were raised about the mRNA level plateau that will need to be addressed, the methods and analyses were generally felt to be sound. Using this clever method, levels of mRNA and protein were used to simultaneously map QTL affecting mRNA and protein expression levels. The primary finding was that different QTL are affecting these two levels of gene expression, which is consistent with prior work. This consistency lead to a range of opinions about the novelty of the work.

In a revision, it will be critical to make more clear the new biological insight provided by the work, beyond the experimental design. Specifically, you should more clearly state what was learned about eQTLs and pQTLs with these new approaches that goes beyond what was already known. Clarifying expectations would also help: Do you expect to simultaneously have the same eQTLs and pQTLs? What are the molecular mechanisms responsible for different mRNA and pQTLs? Authors claim that is not statistical, however large effect trans-QTLs are conserved across e-pQTLs. Either way, how different trans-factors can interact with target genes and be responsible for expression and protein differences?

We have left the full set of reviewer comments below in this case because they contain detailed, specific, and complementary comments and reflect this range of opinions.

Reviewer #1:

In this paper, Brion et al. performed expression QTL mapping in a budding yeast cross using two methods for quantifying gene expression at the level of transcript abundance and protein abundance. Previous eQTL studies have been performed separately for transcript abundance and protein abundance using separate experimental procedures. These studies have found that there is a largely unique set of trans QTL that underlie interindividual variation in mRNA and protein abundance. However, these comparative analyses have a number of limitations that the authors outline. To rigorously address the extent to which these loci overlap the authors developed a method that enables simultaneous detection of protein and mRNA levels in single cells. Protein expression is detected using GFP fusions. mRNA expression is quantified using a novel, and clever, CRISPR-based system that uses self-cleaving ribozymes to generate a guide RNA that directs a dCas9 to drive expression of mCherry. Using this approach the authors performed bulk segregant mapping in a cross between two yeast strains to map loci that underlie differences in expression of mRNA and protein. Consistent with prior studies, they find little overlap between eQTL affecting the two levels of expression. They follow up on one locus, YAK1, in detail using RNAseq and mass spec.

This is an interesting study that presents a very clever method of quantifying mRNA and protein expression simultaneously. The experiments and analysis are well-performed. Although the result is consistent with previous studies, it greatly strengthens those observations and thus is an important addition to the field. Prior to publication the authors should address the following:

– The authors state that "most genetic variation in gene expression arises from trans-acting variants…" and cite several studies from model organisms that support this. However, my understanding is that this is not the case in human studies. The authors should expand on their explanation as to why they haven't been found in humans, beyond the fact that they can occur anywhere in the genome.

– The authors refer to the poor correlation between mRNA and protein levels, but should mention the issues identified with this comparison in Csardi et al., 2015 https://pubmed.ncbi.nlm.nih.gov/25950722/

– Figure 1 states the cells were grown in YNB + glutamate. Why was this media used for the study?

– Figure 3—figure supplement 3 aims to show the heritability of the phenotype of sorted cells. This would be more convincing if scatter plots were presented that show that after the sorting there is no correlation between the green and red signal, or if there is, an explanation as to why.

– It would be easier to read the paper if after the method has been introduced the text referred to mRNA and protein expression rather than the color of the reporter.

– The authors refer to the protein QTL as "post-transcriptional", but I think that term makes one think of mRNA stability, which is not what is measured. Perhaps "translational control" would be more correct and clearer.

– What is the rationale for showing δ-allele frequencies in Figure 4A? Showing LOD scores in this figure would make more sense to me for reporting QTL.

– It is not intuitive to me that δ-allele frequencies correspond to effect sizes and this should be more clearly explained.

– It is not clear to me if there is any compensation used in the FACS analysis. Is there any correlation between the green and red channels when controls containing only green or only red are used?

– Why is a loess regression used to correct for cell size? Wouldn't dividing by FSC be simpler and have the desired effect of normalizing for cell size?

Reviewer #2:

In this work, the authors used a novel approach to simultaneously measure mRNA and protein levels and identify trans-acting genetic variants affecting both or either of both. The authors present a fast and accurate novel strategy, taking advantage of a CRISPR strategy, and FACS analysis in a highly used yeast cross. Unfortunately, this work does not provide novel genetic, neither biological insights and conclusions are similar to previous findings. I would recommend the authors try to extend their work to a higher number of genes/environments and obtained novel biological insights that could explain how trans-acting factors, including post-transcriptional and post-translation modifications, impact mRNA, protein and traits. Furthermore, a time course would be interesting to analyse and determine how trans-acting variants affect mRNA and protein levels over time and whether they correlate or not.

Reviewer #3:

In this manuscript, Brion Lutz and Albert detail the simultaneous mapping of trans-acting QTLs influencing mRNA production and steady-state protein level differences between yeast strains BY4741 and RM-11 for ten genes. While much work has previously been done evaluating mRNA-QTLs in this system and others, the simultaneous comparison of transcription and protein QTLs, even for a relatively small set of genes, is significant and timely. The authors used a clever experimental system that converts transcription rates of test genes into mCherry expression by CRISPR activation. Notably, the authors find most QTLs affect protein levels, but not mRNA levels, and that mRNA-QTLs do not always have the same effects on protein levels. They also report genome-wide comparisons of a YAK1 kinase allele-swap on steady-state protein and mRNA levels. Overall, the manuscript describes important results that suggest more study of pQTLs is needed to better model the impact of transcription regulatory variation. While generally solid, the authors should further evaluate the effects of the mCherry plateau and examine protein-protein interactions for each of the ten genes to evaluate how these might contribute to the disparities between RNA-QTLs and protein-QTLs. They are also missing important citations that should be included for scholarly accuracy. Numbered issues (chronological):

1) Introduction: the authors should include Emerson et al., 2010 for yeast, and references of similar work in *Drosophila*, including McManus et al., 2010; Coolon et al., 2014 and Huang et al., 2015 and potentially others.

2) Introduction paragraph three: Cenik et al., 2015 also used ribosome profiling and mass spectrometry to examine post-transcriptional variation amont human LCLs and should be cited.

3) One caveat to the experimental approach is the plateau in mCherry measurements at high transcription rates. The authors argue that this plateau does not affect most genes, by comparing qPCR signal from inducible GFP and ACT1. However, this seems problematic because the PCR efficiency of GFP and ACT1 may not be identical. Thus it's possible the mRNA plateau occurs at a lower level. If this were the case, one would expect to see fewer mRNA-QTLs for highly expressed genes than for low-expression genes. Is there any relationship between expression levels (RNA-seq) and number of mRNA-QTLs among the 10 genes tested? If high mRNA genes are artifactually missing mRNA-QTLs, it's possible more pQTLs also affect mRNA levels.

4) A related complication of the assay system is that the tested genes do not have native 3' UTRs or cleavage and polyadenylation sites (CPS). The "strength" of CPS, has an important impact on mRNA levels and may influence both mRNA stability and transcription (See Shalem et al. PMID: 25875337). The 3' UTR issue should be discussed.

5) "A majority of the loci corresponded to protein-QTLs that did not overlap an mRNA-QTL". Were these non-matching QTLs more common in genes with high mRNA levels (i.e. could this reflect the mCherry plateau)?

6) Figure 5 shows variation among the number of pQTLs and mRNA-QTLs for the 10 studied genes. In the Discussion, the authors note that some p-QTLs could result by changes in protein complex members. Is there any relationship between the number of pQTLs for genes and the number of protein-protein interactions each gene has?

7) The YAK1 mutation was identified as altering protein (but not mRNA) levels of ARO8, BMH2 and GPD1. The genome-wide RNA-seq and Mass-spec comparison of YAK1 alleles appears to only show this affect for GPD1. Were the other genes also significantly altered in mRNA or protein levels in this orthogonal experiment?

8) The paper ends with the suggestion that protein abundance is under more complex genetic control than mRNA abundance. While this seems very likely in general (before the results presented in this paper) due to protein turnover and post-translational modifications, I think the authors should reiterate at the end of the paper that this is for trans-QTLs specifically. I expect that cis-QTLs have more consistent influences on mRNA and protein levels.

---

## [Author Response]

This paper presents a creative, novel strategy for quantifying expression of RNA and protein of the same genes in the same cells. While some concerns were raised about the mRNA level plateau that will need to be addressed, the methods and analyses were generally felt to be sound. Using this clever method, levels of mRNA and protein were used to simultaneously map QTL affecting mRNA and protein expression levels. The primary finding was that different QTL are affecting these two levels of gene expression, which is consistent with prior work. This consistency lead to a range of opinions about the novelty of the work.In a revision, it will be critical to make more clear the new biological insight provided by the work, beyond the experimental design. Specifically, you should more clearly state what was learned about eQTLs and pQTLs with these new approaches that goes beyond what was already known. Clarifying expectations would also help: Do you expect to simultaneously have the same eQTLs and pQTLs? What are the molecular mechanisms responsible for different mRNA and pQTLs? Authors claim that is not statistical, however large effect trans-QTLs are conserved across e-pQTLs. Either way, how different trans-factors can interact with target genes and be responsible for expression and protein differences?

1) The plateau in mRNA levels has been addressed as follows:

– As requested by reviewer #2, we examined the robustness of our chosen threshold of half the abundance of *ACT1* mRNA by comparing the abundance of *ACT1* we had used to that in another RNA-Seq dataset. In both datasets, the expression of the vast majority of genes are below this threshold, which is consistent with our prior interpretation of these data in the paper. We have also reworded the manuscript to make clear that we view this “threshold” as a broad guideline rather than a hard, numerically precise threshold.

– As requested by reviewer #3, we asked whether mRNA abundance as measured by external RNA-Seq data correlates with the number of mRNA-QTLs or protein-specific QTLs we detected. We found no such relationship, suggesting that the mCherry plateau does not greatly influence our results.

2) Our manuscript presents several novel advances:

First, and most obviously, we introduce a novel method. We anticipate that the reporter system we describe will find uses well beyond QTL mapping. To name just one example, at conferences where we have presented this work, we had discussions with bioengineers who were considering our method for real-time monitoring of the production of compounds or peptides in yeast bioreactors, in cases where large protein tags such as GFP would interfere with compound production. While detailed discussion of these possibilities is out of the scope of the current manuscript, we are confident that our method will be useful in many applications.

Regarding a perceived lack of biological insight, we believe that this was largely an issue with our previous presentation of the background literature. We have now addressed this with a completely rewritten Introduction (see point 3 below) to highlight the lack of consensus in the field on the relationship between eQTLs and pQTLs. We started this study believing this lack of consensus is mostly due to technical limitations of prior work (small sample size and experimental variability). We explicitly designed our system to be as free from these issues as possible. Thus, a possible outcome of our work was that in our well-controlled system, most discrepancies between eQTLs and pQTLs would disappear. The fact that these discrepancies did not disappear is an important and novel result.

Finally, we want to stress that our manuscript includes fine-mapping of a *trans*-acting pQTL to the causal variant. Fine-mapping causal variants is not a trivial task, making this fine-mapped variant a novel result in its own right.

3) To help readers appreciate the novelty of our results, we have followed the editor’s suggestion to clarify expectations regarding the discrepancies between eQTLs and pQTLs. To do so, we rewrote the Introduction entirely. It now includes an expanded review of the eQTL/pQTL literature in multiple species (human, mouse, plants, and yeast). The new Introduction details discrepancies between the results reported by different studies. Briefly, different studies report different degrees to which a typical eQTL is also a pQTL for the same gene. There is also a range of reports on what fraction of pQTLs arises from protein-specific processes. We see no consensus in the literature on these questions, and our Introduction now makes this point more forcefully.

We go on to describe in more detail than before exactly how the technical shortcomings of earlier work (i.e., small sample size and / or comparisons across studies) may have inflated discrepancies between eQTLs and pQTLs. We believe that this rewritten Introduction makes it clear why the work we describe here was necessary, and why our results matter: although our system controlled for environmental factors and had high statistical power, we still found many discrepancies between eQTLs and pQTLs.

4) Regarding the comment “Authors claim that [it] is not statistical, however large effect trans-QTLs are conserved across e-pQTLs.”: This is a fair point, and we discuss this below in more detail in our response to reviewer #2. To briefly restate here, we cannot formally rule out that incomplete power to detect mRNA-QTLs may have contributed to some apparent protein-specific QTLs, just like we cannot rule out the inverse scenario for mRNA-specific QTLs. Our manuscript explicitly acknowledges this possibility, includes analyses that aim to address limited statistical power, and provides evidence that while incomplete power undoubtedly leads to some false calls of specific QTLs, it is unlikely to account for all of them.

5) Regarding the comments “the molecular mechanisms responsible for different mRNA and pQTLs” and “how different trans-factors can interact with target genes and be responsible for expression and protein differences?”. To our knowledge, the only protein-specific *trans*-pQTLs that have been reported in some detail are effects on protein complex stoichiometry (although we are not aware of experimental confirmation of these causal relationships). This mechanism is discussed in the revised Discussion section. We agree that experimental dissection of additional protein-specific *trans*-pQTLs would help to elucidate the mechanistic nature of eQTL and pQTL; however, we believe this is out the scope of the present paper, given the considerable challenges in identifying the causal genes and variants in QTL regions.

Reviewer #1:[…]This is an interesting study that presents a very clever method of quantifying mRNA and protein expression simultaneously. The experiments and analysis are well-performed. Although the result is consistent with previous studies, it greatly strengthens those observations and thus is an important addition to the field. Prior to publication the authors should address the following:– The authors state that "most genetic variation in gene expression arises from trans-acting variants…" and cite several studies from model organisms that support this. However, my understanding is that this is not the case in human studies. The authors should expand on their explanation as to why they haven't been found in humans, beyond the fact that they can occur anywhere in the genome.

It is indeed true that individual *trans*-eQTLs (let alone causal variants in these eQTL regions) are harder to find in human populations than in model organisms. However, as a group, *trans*-eQTLs contribute more genetic variation to gene expression than *cis*-eQTLs in humans, just like in model organisms. These human results are not based on the effects of individual *trans*-eQTLs, but on genome-wide aggregate estimates from mixed linear models. These findings are described in two studies of human populations Grundberg et al., and Wright et al., which we cite in our Introduction.

To clarify these points, we have reworded this section as part of our extensively rewritten Introduction.

– The authors refer to the poor correlation between mRNA and protein levels, but should mention the issues identified with this comparison in Csardi et al., 2015 https://pubmed.ncbi.nlm.nih.gov/25950722/

We have added this reference as part of the rewritten Introduction.

– Figure 1 states the cells were grown in YNB + glutamate. Why was this media used for the study?

This medium was used only in the experiments in Figure 1. We had conducted these experiments in the presence of G418 (which requires the use of YNB + glutamate) because, at the time, we were concerned about potential loss of the reporter construct from the genome due to recombination between its flanking sequences. This concern later turned out to be unfounded. In the remainder of this study, we used synthetic complete medium, as indicated in the Materials and methods and the figure legends throughout the paper. We have also added more explanation for our choice of media to the Materials and methods section.

– Figure 3—figure supplement 3 aims to show the heritability of the phenotype of sorted cells. This would be more convincing if scatter plots were presented that show that after the sorting there is no correlation between the green and red signal, or if there is, an explanation as to why.

This is an interesting point. First, we’d like to emphasize that these experiments, in which we measured the response to phenotypic selection after a few generations of growth, are informed by the breeder’s equation in quantitative genetics. This equation relates the strength of selection that is applied to a population (defined as the difference in mean phenotype between selected and unselected individuals) to the phenotypic response to selection (defined as the shift in mean of the population after one generation). The breeder’s equation is usually applied to a single generation of sexually reproducing diploid organisms, making its use in our haploid segregants (which undergo multiple mitotic cell divisions but no meiosis after FACS) imperfect. Nevertheless, it provides a framework for interpreting the shifts in mean phenotypes that we reported.

Second, our selection scheme would not necessarily be expected to result in a strong reduction in correlation between mRNA and protein. Specifically, the effects of each individual locus are small compared to the nongenetic cell-to-cell noise in gene expression, which can reach up to 100-fold variation even in isogenic yeast cell populations. This means that our selections are most likely not strong enough to result in pure populations, in which every cell carries all alleles that increase (or decrease) protein or mRNA at all or even most loci across the genome. Instead, we fully expect that our sorted extreme populations still contain alleles for high and low expression at many QTLs. This expectation is confirmed by the fact that complete fixation of opposite alleles in high vs low populations was almost never observed in this work, consistent with our previous bulk-segregant studies.

Therefore, each of our extreme populations still segregates at multiple QTLs. Because a portion of these QTLs do influence both mRNA and protein, these loci are expected to create a correlation between mRNA and protein even after selection. To confirm this, we computed the correlations between red and green fluorescence in each population, and compared their magnitude before and after sorting:

In Author response image 1, the y-axis shows the difference between the mCherry/GFP correlation coefficients measured after FACS and growth of an unsorted population (“all_pop”) or the fluorescence-based sorted population (“sorted_pop”). The numbered populations are: 1=low mCherry, 2=high mCherry, 3=low GFP, 4=high GFP. There is one subplot per gene, with lines connecting the four populations from a given biological replicate. The “NA” population refers to untagged cells (as explained in Figure 3—figure supplement 3).

**Author response image 1. sa2fig1:** 

In most populations, there is not a consistent difference in correlation coefficients between cells during FACS and after growth. The only consistent exception is the gene *CYC1*, where the post-growth cells do show reduced correlation. This gene is affected by a strong QTL, such that FACS can create relatively “pure” populations of cells that all share one of the alleles at this locus. Note that even here, the other QTLs affecting this gene do still segregate and create correlations among single cells. We hope that these explanations address this comment. We have chosen to keep our presentation of the results in Figure 3—figure supplement 3 unchanged.

– It would be easier to read the paper if after the method has been introduced the text referred to mRNA and protein expression rather than the color of the reporter.

This is a great point, and one that we had discussed extensively while preparing our manuscript. As the reviewer will appreciate, there is a tradeoff here between slightly greater ease of reading and more exact language. Specifically, while we certainly assume (and believe to have shown) that “red” mCherry corresponds to mRNA and “green” GFP to protein, we ultimately can only say with certainty that we measure fluorescence signals during flow cytometry. Therefore, when describing flow cytometry, we had opted to refer to the observed quantities (i.e. fluorescence) rather than the molecules they report on. For example, we do feel it is more appropriate to say that sorting was performed based on “mCherry”, rather than, say “mRNA”. However, note that as the text progresses to presenting the QTL results, it already has a transition in terminology to “mRNA-QTLs” and “protein-QTLs”, as opposed to “mCherry-QTLs” and “GFP-QTLs”.

We have reworded some instances of where this terminology may have been confusing (e.g. Figure 4 legend).

– The authors refer to the protein QTL as "post-transcriptional", but I think that term makes one think of mRNA stability, which is not what is measured. Perhaps "translational control" would be more correct and clearer.

Our use of “post-transcriptional” was intended to be understood very literally, to refer to any process that may affect protein abundance after transcription. As the reviewer correctly notes, because our mCherry system stops reporting on the fate of the mRNA after the gRNA has been released, it indeed cannot report on mRNA degradation (other than for some mRNAs that are degraded before gRNA release is complete). However, GFP-tag signal is sensitive to mRNA degradation, to the extent that mRNA degradation limits the time available for translation and therefore the number of protein molecules that can be generated from a given mRNA molecule. Further, we feel that “translational control” is too specific, since it omits possible influences of protein degradation on protein abundance. For these reasons, we think that the term “post-transcriptional” is appropriately specific [because it only excludes the one process (transcription) that will have the strongest influence on mCherry signal] but also sufficiently broad (because it captures any mechanism that can shape protein abundance downstream of transcription). Therefore, we have decided to keep this term in the paper. We have added a brief statement to the Discussion that makes clear that we take “post-transcriptional” to mean “mRNA stability, translation, or protein degradation”

– What is the rationale for showing δ-allele frequencies in Figure 4A? Showing LOD scores in this figure would make more sense to me for reporting QTL.

The rationale for showing δ-AF is four-fold. First, LOD scores would not show the direction of effect at a given QTL (i.e., whether the BY or the RM allele increases expression), while δ-AF does show this. Second, the relationship between δ-AF and LOD scores is not linear: as δ-AF approaches fixation, the LOD score rises quickly. From a visualization standpoint, this creates the issue that strong QTLs, if shown at full scale, “blow out” the display, such that weaker loci become very hard to see. These two points are the main reason we chose to use δ-AF in this figure.

Third, LOD scores estimated by the “Multipool” algorithm we use are sensitive to the depth of sequencing coverage. At higher coverage, LOD scores increase for a given effects size, because deeper sequencing provides more certainty to the model that the allele frequencies in the two populations are not the same. We acknowledge that this is not a major concern for our present work, because sequencing coverage is fairly similar across our samples. Fourth, the δ-AFs are a more “raw” display of the data than LOD scores; these allele frequencies are directly derived from the actual measurements (i.e. allele counts in Illumina data). Together, these points led us to use δ-AF as the more appropriate choice for displaying the data visually.

Note that our supplementary tables do of course report LOD scores, along with many other characteristics of each QTL, such that interested readers have full access to this information.

– It is not intuitive to me that δ-allele frequencies correspond to effect sizes and this should be more clearly explained.

We illustrate the relationship between effect size and δ AF in Author response image 2, which is based on simulated bulk segregant mapping experiments. We have also added a brief explanation of these considerations to the Materials and methods, at the end of the “QTL mapping” section.

Consider a population of cells made up of two subpopulations that each carry one of two alleles at a QTL that affects the given trait (left panel in Author response image 2). The effect of the QTL is given by the difference in mean between the two subpopulations. When we use FACS on the combined population, we use the indicated thresholds for picking “high” vs “low” cells. Note that in the high population defined by the “high” threshold, the fraction of cells with the BY allele is lower than the cells with the RM allele, and vice versa for the low population. The ratio of BY to RM alleles in the extreme populations determines the allele frequencies, and subtracting these from each other gives the δ AF metric that we use as a measure of effect size.

If we increase the effect size of the QTL, the two subpopulations are being pushed further apart, and the degree of genotype enrichment in the two extreme populations increases. This increases δ AF, as shown on the right for a range of simulated effect sizes. Note that the relationship between effect size and δ AF is not linear, especially for strong QTLs, but that δ AF rises monotonically as a function of effect size.

– It is not clear to me if there is any compensation used in the FACS analysis. Is there any correlation between the green and red channels when controls containing only green or only red are used?

We did not use compensation during FACS. This information has been added to the Materials and methods. During our pilot studies for this work, we did not notice interference between the green and red fluorescence signals.

– Why is a loess regression used to correct for cell size? Wouldn't dividing by FSC be simpler and have the desired effect of normalizing for cell size?

We chose LOESS regression over simple division, because LOESS allows for relationships between FSC and fluorescence that are not strictly linear with a constant slope across the entire range of the data. In practice, there is almost no difference between correlations between red and green fluorescence using LOESS vs division, see Author response image 3.

**Author response image 3. sa2fig3:** 

Reviewer #2:In this work, the authors used a novel approach to simultaneously measure mRNA and protein levels and identify trans-acting genetic variants affecting both or either of both. The authors present a fast and accurate novel strategy, taking advantage of a CRISPR strategy, and FACS analysis in a highly used yeast cross. Unfortunately, this work does not provide novel genetic, neither biological insights and conclusions are similar to previous findings. I would recommend the authors try to extend their work to a higher number of genes/environments and obtained novel biological insights that could explain how trans-acting factors, including post-transcriptional and post-translation modifications, impact mRNA, protein and traits. Furthermore, a time course would be interesting to analyse and determine how trans-acting variants affect mRNA and protein levels over time and whether they correlate or not.

We thank the reviewer for their positive assessment of our method. While we look forward to applying our reporter system to additional proteins and questions in future work, we consider the scale of experiments suggested by the reviewer to be out of the scope of the present manuscript. In addition, we politely disagree that our work has not provided novel insights, but do acknowledge that our presentation of the background literature for this work may have obscured some novelty, as also pointed out by the editor.

To address this, we have worked to improve our presentation of the background literature as well as our new insights in the paper. Specifically, we have rewritten the entire Introduction to provide a more detailed review of existing comparisons between eQTLs and pQTLs from several species (human, mouse, plants, and, of course, yeast). In our opinion, the literature is far from settled on how well eQTLs and pQTLs agree. For example, while some studies have shown that most *cis*-acting eQTLs are also pQTLs, others have reported systematic buffering of the effects of *cis*-eQTLs such that they have smaller, if any, effects on the proteins of the same gene. Likewise, for *trans*-eQTLs, some studies report nearly complete lack of overlap (indeed, Foss et al., 2011 even pointed out that *trans*-QTLs may show less overlap than expected by chance), while others focus on relatively strong correlations of effect sizes to indicate that most *trans*-acting variants have consistent effects. Whether or not eQTL/pQTL discrepancies exist is an important and open question. Among other implications for the biology of gene expression, this question raises the practical issue of how informative existing studies of mRNA expression variation are, including many large-scale projects in human genetics, if mRNA expression changes do not correlate with changes in protein expression.

Our revised Introduction now stresses more explicitly technical considerations in earlier work related to 1) low statistical power and 2) experimental differences between studies across which eQTLs and pQTL are almost always compared. These two issues have the potential to create the impression of discrepant eQTLs and pQTLs, even if eQTLs and pQTLs were consistent in reality.

Our new approach here was explicitly designed to directly address both of these issues. We aimed to create a system that maximizes the chance of detecting consistency between eQTLs and pQTLs. The fact that we nevertheless detected considerable discrepancies between eQTLs and pQTLs is an important result. In addition to our rewritten Introduction, we have also emphasized these points more in the Discussion section.

On a more technical level, we are convinced that our reporter system could find uses outside of mapping regulatory genetic variation. Finally, we stress that fine-mapping QTLs to the causal variant remains a difficult challenge. That we were able to accomplish it here for the *YAK1* gene adds further novelty to our study.

Reviewer #3:In this manuscript, Brion Lutz and Albert detail the simultaneous mapping of trans-acting QTLs influencing mRNA production and steady-state protein level differences between yeast strains BY4741 and RM-11 for ten genes. While much work has previously been done evaluating mRNA-QTLs in this system and others, the simultaneous comparison of transcription and protein QTLs, even for a relatively small set of genes, is significant and timely. The authors used a clever experimental system that converts transcription rates of test genes into mCherry expression by CRISPR activation. Notably, the authors find most QTLs affect protein levels, but not mRNA levels, and that mRNA-QTLs do not always have the same effects on protein levels. They also report genome-wide comparisons of a YAK1 kinase allele-swap on steady-state protein and mRNA levels. Overall, the manuscript describes important results that suggest more study of pQTLs is needed to better model the impact of transcription regulatory variation. While generally solid, the authors should further evaluate the effects of the mCherry plateau and examine protein-protein interactions for each of the ten genes to evaluate how these might contribute to the disparities between RNA-QTLs and protein-QTLs. They are also missing important citations that should be included for scholarly accuracy. Numbered issues (chronological):1) Introduction: the authors should include Emerson et al., 2010 for yeast, and references of similar work in Drosophila, including McManus et al., 2010; Coolon et al., 2014 and Huang et al., 2015 and potentially others.

We thank the reviewer for these suggestions. As part of our revised Introduction, we have added the requested references, along with others. Some of the references suggested by the reviewer examined F1 hybrids and parent strains, but did not perform linkage analysis in segregants (which was the intended topic of this sentence) and in some cases examined divergent species rather than strains of the same species. For these reasons, we had not cited these papers before. To accommodate these references in the revision, we have changed the sentence in the manuscript as follows:

“By contrast, experimental crosses in organisms such as yeast (Albert et al., 2018; Brem et al., 2002; Brion et al., 2020; Clément-Ziza et al., 2014; Emerson et al., 2010; Thompson and Cubillos, 2017), plants (Fu et al., 2013; West et al., 2007; Zhang et al., 2011), worms (Snoek et al., 2017; Viñuela et al., 2010), flies (Coolon et al., 2014; Everett et al., 2020; Huang et al., 2015; McManus et al., 2010), and mouse (Gerrits et al., 2009; Hasin-Brumshtein et al., 2016) have quantified the global contribution of cis and trans-acting variation and identified thousands of trans-eQTLs.”

2) Introduction paragraph three: Cenik et al., 2015 also used ribosome profiling and mass spectrometry to examine post-transcriptional variation amont human LCLs and should be cited.

We have added this reference, as well as Wu et al., 2013 as an early application for mass spectrometry for pQTL mapping in human cells.

3) One caveat to the experimental approach is the plateau in mCherry measurements at high transcription rates. The authors argue that this plateau does not affect most genes, by comparing qPCR signal from inducible GFP and ACT1. However, this seems problematic because the PCR efficiency of GFP and ACT1 may not be identical. Thus it's possible the mRNA plateau occurs at a lower level. If this were the case, one would expect to see fewer mRNA-QTLs for highly expressed genes than for low-expression genes. Is there any relationship between expression levels (RNA-seq) and number of mRNA-QTLs among the 10 genes tested? If high mRNA genes are artifactually missing mRNA-QTLs, it's possible more pQTLs also affect mRNA levels.

We have conducted the analysis reviewer suggested, and found no relationship between a gene’s mRNA expression in the Albert, Bloom et al., 2018 data and the number of mRNA-QTLs (or any other type of QTL). Specifically, while the correlation between the number mRNA-QTLs and mRNA abundance did have a negative coefficient (rho = -0.15), the correlation was not significant (p = 0.67). Thus, this analysis suggests that the mCherry plateau does not cause systematic underestimation of the number of mRNA-QTLs in our data.

However, we agree with the reviewer’s point that the comparison of *ACT1* qPCR and RNA-seq is not without problems, as also noted by reviewer 2 above. This is true even though we did correct for different PCR efficiencies using the calibration curves shown in Figure 1—figure supplement 2.

In the revised manuscript, we explicitly acknowledge this issue and have softened our language describing how many genes can likely be analyzed with our reporter system.

4) A related complication of the assay system is that the tested genes do not have native 3' UTRs or cleavage and polyadenylation sites (CPS). The "strength" of CPS, has an important impact on mRNA levels and may influence both mRNA stability and transcription (See Shalem et al. PMID: 25875337). The 3' UTR issue should be discussed.

We agree, and have added a discussion of the possible consequence of 3’ UTR removal to the Discussion section:

“Like other reporters that tag the 3’ end of genes, our system decouples the native 3’ UTR from the gene of interest. 3’ UTRs can influence gene expression, and genetic trans-effects that involve mechanisms that target the 3’ UTR would be missed by our system. However, loss of the native 3’ UTR is not expected to inflate differences between mRNA-QTLs and protein-QTLs. Instead, 3’ UTR removal may reduce differences between mRNA-QTLs and protein-QTLs because it eliminates an element of the mRNA that could be a target for mRNA-specific processes”

5) "A majority of the loci corresponded to protein-QTLs that did not overlap an mRNA-QTL". Were these non-matching QTLs more common in genes with high mRNA levels (i.e. could this reflect the mCherry plateau)?

Across the ten genes, the correlation between the number of protein-specific QTLs and mRNA abundance had a rho value of -0.02 with a p-value of 0.96. We do not think that the mCherry plateau is responsible for these protein-specific QTLs.

6) Figure 5 shows variation among the number of pQTLs and mRNA-QTLs for the 10 studied genes. In the Discussion, the authors note that some p-QTLs could result by changes in protein complex members. Is there any relationship between the number of pQTLs for genes and the number of protein-protein interactions each gene has?

To address this question, we extracted physical protein/protein interactions from the bioGRID database (https://thebiogrid.org/). Genes with more of these interactions tended to have a higher number of protein-QTLs (rho = 0.51) as well as protein-specific QTLs (rho = 0.55) but neither of these results was statistically significant (p-values: 0.13 and 0.10, respectively). We suspect that ten genes is likely not enough to test this question rigorously. Future analyses with more genes may well strengthen the significance of this relationship. We thank the reviewer for this excellent suggestion, which we will keep in mind for future work.

7) The YAK1 mutation was identified as altering protein (but not mRNA) levels of ARO8, BMH2 and GPD1. The genome-wide RNA-seq and Mass-spec comparison of YAK1 alleles appears to only show this affect for GPD1. Were the other genes also significantly altered in mRNA or protein levels in this orthogonal experiment?

The results are shown in Author response table 1, and can also be found in the source data for Figure 5—figure supplement 2 and Figure 7—figure supplement 3:

**Author response table 1. resptable1:** 

Gene	mCherry deltaAF (LOD)	RNASeq log2(Fold change)	RNASeq Raw P-value (adjusted p-value)	GFP deltaAF (LOD)	Mass-spec log2(Fold change)	Mass-spec Raw P-value (adjusted p-value)
*GPD1*	0.02 (0.6)	-0.12	0.01 (0.10)	-0.4 (94.2)	-0.17	0.0001 (0.004)
*ARO8*	0.05 (1.8)	0.05	0.24 (0.55)	-0.1 (6.7)	0.13	0.09 (0.4)
*BMH2*	0.04 (1.0)	0.03	0.21 (0.51)	-0.20 (22.2)	-0.01	0.93 (0.98)

As the reviewer correctly spotted, *ARO8* and *BMH2* were not significantly different in either RNA-seq or mass spectrometry. We believe that this is likely due to the small effect of the *YAK1* variant on the protein levels of these two genes, which is much smaller than its effect on Gpd1. In Author response table 1, note the big differences among these genes in terms of their deltaAF metric (which is related to the effect size of a QTL; see our response to reviewer 1 above) as well as their LOD scores, which show that *GPD1* has a much stronger protein-QTL than the other two genes. Our mass-spectrometry dataset with n = 5 in each group is likely not powered to detect the small effects at the other two genes.

8) The paper ends with the suggestion that protein abundance is under more complex genetic control than mRNA abundance. While this seems very likely in general (before the results presented in this paper) due to protein turnover and post-translational modifications, I think the authors should reiterate at the end of the paper that this is for trans-QTLs specifically. I expect that cis-QTLs have more consistent influences on mRNA and protein levels.

We have clarified in the final paragraph of the paper that our results exclusively cover *trans*-QTLs.